# SGD with a Constant Large Learning Rate Can Converge to Local Maxima

**Liu Ziyin**[1]**, Botao Li**[2]**, James Simon**[3]**, & Masahito Ueda**[1]
[1]The University of Tokyo
[2]ENS, Université PSL, CNRS, Sorbonne Université, Université de Paris
[3]University of California, Berkeley

## ABSTRACT

Previous works on stochastic gradient descent (SGD) often focus on its success. In this work, we construct worst-case optimization problems illustrating that, when not in the regimes that the previous works often assume, SGD can exhibit many strange and potentially undesirable behaviors. Specifically, we construct landscapes and data distributions such that (1) SGD converges to local maxima, (2) SGD escapes saddle points arbitrarily slowly, (3) SGD prefers sharp minima over flat ones, and (4) AMSGrad converges to local maxima. We also realize results in a minimal neural network-like example. Our results highlight the importance of simultaneously analyzing the minibatch sampling, discrete-time updates rules, and realistic landscapes to understand the role of SGD in deep learning.

## 1 INTRODUCTION

SGD is the main optimization algorithm for training deep neural networks, and understanding SGD is widely regarded as a key step on the path to understanding deep learning (Bottou, 2012; Zhang et al., 2018; Xing et al., 2018; Mori et al., 2021; Du et al., 2018; Allen-Zhu et al., 2018; Wojtowytsch, 2021a;b; Gower et al., 2021; Ziyin et al., 2022b; Gurbuzbalaban et al., 2021; Zou et al., 2021; Li et al., 2021; Feng and Tu, 2021). Despite its algorithmic simplicity (could be described by only two lines of equations), SGD is hard to understand. The difficulty is threefold: (1) SGD is discrete-time in nature, and discrete-time dynamics are typically much more complicated to solve than their continuous-time counterparts (May, 1976); (2) SGD noise is state-dependent (Ziyin et al., 2022b; Hodgkinson and Mahoney, 2020); and (3) the loss landscape can be nonlinear and non-convex, involving many local minima, saddle points, and degeneracies (the Hessian is not full-rank) in the landscape (Xing et al., 2018; Wu et al., 2017). Each of these difficulties is so challenging that very few works attempt to deal with all of them simultaneously. Most previous works on SGD are limited to the cases when the loss landscape is strongly convex (Ziyin et al., 2022b; Liu et al., 2021; Hoffer et al., 2017; Mandt et al., 2017), or when the noise is assumed to be Gaussian and time-independent (Zhu et al., 2019; Jastrzebski et al., 2017; Xie et al., 2021); for the works that tackle SGD in a non-convex setting, often strong conditions are assumed. The reliance on strong assumptions regarding each of the challenges means that our present understanding of SGD for deep learning could be speculative. This work aims to examine some commonly held presuppositions about SGD and show that when all the three challenging factors are taken together, many counter-intuitive phenomena may arise. Since most of these phenomena are potentially undesired, this work can also be seen as a worst-case analysis of SGD.

In this work, we study the behavior of SGD in toy landscapes with non-convex loss and multi-minima. We approach the problem of SGD convergence from a different angle from many of the related works: instead of studying when SGD will converge, our result helps answer the question of when SGD might fail. In particular, the problem setting considers discrete-time SGD close to saddle points, where the noise is due to minibatch sampling, and the learning rate is held constant throughout training. In the next section, we define the SGD algorithm and the necessary notations. In Sec. 3, we discuss the related works. A warmup example is provided in Sec. 4. Sec. 5 introduces the main results. Sec. 6 presents the numerical simulations, including a minimal example involving a neural network. We also encourage the readers to examine the appendix. Sec. A presents additional experiments. Sec. B

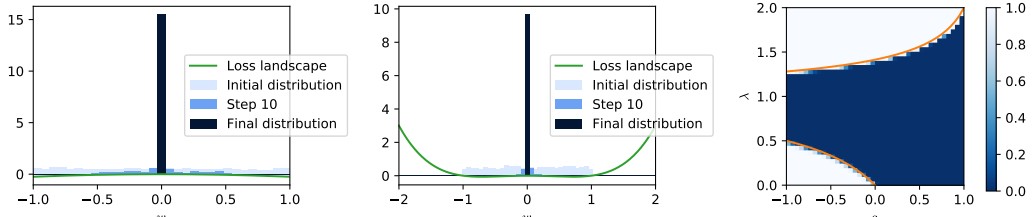

Figure 1: SGD converges to a local maximum for the example studied in Sec. 5.1. **Left**: Distribution of $w$ on the quadratic potential, $\hat{L} = xw^2/2$; the histograms in different colors show the distribution of model parameters at different time steps. **Middle**: Distribution on the fourth-order potential, $\hat{L} = xw^2/2 + w^4/4$. **Right**: Escape probability as a function of $a$ and $\lambda$. The parameters space is divided into an absorbing phase where $w$ is attracted to the local maximum (in **dark blue**) and an active phase where $w$ successfully escapes the two central bins (in white).

presents a continuous-time analysis of the problems in the main text and is relevant for discussing the unique discrete-time features of SGD. Sec. C presents all the proofs.

## 2 BACKGROUND

**Notation and Terminology**. We use $\lambda$ to denote the learning rate. $w$ is the model parameter. $\hat{L}(w; x)$, with the caret symbol $\hat{}$, denotes the *sampled* loss function, which is a function of the data point, and $L := \mathbb{E}_x[\hat{L}]$ is the loss landscape averaged over the data distribution; for notational conciseness, we often hide the dependence of $\hat{L}$ on $x$ when the context is clear. The lowercase $t$ denotes the time step of optimization. The phrases *loss*, *objective*, and *potential* are also used interchangeably.

Now, we introduce the minibatch SGD algorithm. The objective of SGD can be defined as a pair $(\hat{L}, p(x))$ such that $\hat{L}$ is a differentiable function and $p(x)$ is a probability density. We aim at finding the minimizer of the following differentiable objective:

$$L(w) = \mathbb{E}_{x \sim p(x)}[\hat{L}(w; x)], \tag{1}$$

where $x$ is a data point drawn from distribution $p(x)$ and $w \in \mathbb{R}^D$ is the model parameter. The gradient descent (GD) algorithm for $L$ is defined as the update rule $w_t = w_{t-1} - \lambda \nabla_w L(w, x)$ for some randomly initialized $w_0$, where $\lambda$ is the learning rate. The following definition defines SGD.

**Definition 1.** The *minibatch SGD* algorithm computes the update to the parameter $w$ with the following equation: $w_t = w_{t-1} - \lambda \hat{g}_t$, where $\hat{g}_t = \nabla_{w_{t-1}} \hat{L}(w_{t-1}; x_t)$, and $x_t$ is drawn from $p(x)$ such that $x_i$ is independent of $x_j$ for $i \neq j$.

## 3 RELATED WORKS

**Escaping Saddle Points**. Because neural networks are highly nonlinear, the landscape of neural networks is believed to have many local minima and saddle points (Du et al., 2017; Kleinberg et al., 2018; Reddi et al., 2018b). It is thus crucial for the optimization algorithm to be able to overcome saddle points efficiently and escape a suboptimal minimum. However, the majority of works on escaping saddle points assume no stochasticity or only artificially injected noise (Li et al., 2017; Du et al., 2017; Jin et al., 2017; Ge et al., 2015; Reddi et al., 2018b; Lee et al., 2016; Pemantle, 1990). Some physics-inspired approaches take the stochasticity into account but often rely on continuous-time approximation and nontrivial assumptions regarding the stationary distribution of the parameter (Mori et al., 2021; Xie et al., 2021; Liu et al., 2021). In the context of stochastic non-convex optimization, one setting that SGD is known to escape saddles and converge well is when the learning rate decreases to zero in the training (Pemantle, 1990; Vlaski and Sayed, 2019; Mertikopoulos et al., 2020). In contrast, we only consider the case when the learning rate is held fixed. We stress that our result does not contradict these previous results. On the opposite, our result reinforces the idea that the learning rate needs to be carefully chosen and scheduled when convergence is an essential concern.

**Role of Minibatch Noise**. One indispensable aspect of SGD is its stochasticity originated from minibatch sampling (Wu et al., 2020; Ziyin et al., 2022b; Zhu et al., 2019; Hodgkinson and Mahoney,

2020), whose strength and structure are determined by the model architecture and the data distribution. Moreover, it has been observed that the models trained with SGD significantly outperform the models directly trained with GD (Hoffer et al., 2017). Therefore, understanding the role of minibatch noise is of both fundamental and practical value. However, in the previous works, the focus is on analyzing the behavior of SGD on the landscape specified by $L$; the unique structures due to the noise in SGD are often treated in an oversimplified manner without referring to the actual models in deep learning or the actual data distributions. For example, Daneshmand et al. (2018) assumes that the noisy gradient is negatively correlated with the least-eigenvalue direction of the Hessian, an assumption that its validity is unclear when the Hessian is not full rank. Kleinberg et al. (2018) assumes that the noise in SGD makes the landscape one-point strongly convex, which implies that the loss landscape before the convolution is already close to a strongly convex landscape. Another commonly assumed condition is that the loss-landscape satisfies the Lojasiewicz condition, which is non-convex but implies the unrealistic condition that there is no local minimum except the global minimum (Karimi et al., 2020; Wojtowytsch, 2021a; Vaswani et al., 2019), which is not sufficient to understand the complicated dynamics that often happen in a setting with many minima (for example, a deep linear net (Ziyin et al., 2022a)). In Sec. 6.3, we show that these assumptions are violated for a minimal nonlinear neural network with two layers and one hidden neuron. While our example may or may not be directly relevant for realistic settings in the deep learning practice, our work highlights the importance of analyzing the actual noise structure imposed by a deep neural network in future works.

**Large Learning Rate Regime**. Recently, it has been realized that networks trained at a large learning rate have a better performance than networks trained with a vanishing learning rate (lazy training) (Chizat and Bach, 2018). Lewkowycz et al. (2020) shows that there is a qualitative difference between the lazy training regime and the large learning rate regime; the performance features two plateaus in testing accuracy in the two regimes, with the large learning rate regime performing much better. However, almost no previous theory exists for understanding SGD at a large learning rate. Our work is also relevant for understanding what happens at a large learning rate.

## 4    A WARM-UP EXAMPLE

This section studies a special example of the main results of our work. While the setting of this example is restricted, it captures the essential features and mechanisms of SGD that we will utilize to prove the main results. Consider the following loss function:

$$\hat{L}(w) = \frac{x}{2}w^2, \tag{2}$$

where $x, w \in \mathbb{R}$ and $x$ is drawn from an underlying distribution $p(x)$ at every time-step. We let $\mathbb{E}[x] = a/2$ and assume that $\mathrm{Var}[x] = \sigma^2$ is finite. This gives rise to a true, sample-averaged loss of $L(w) = \frac{a}{4}w^2$. The stationary point of this loss function is $w = 0$. When $a > 0$, the underlying (deterministic) landscape is a minimum; the $a > 0$ case is now well understood in the discrete-time limit and when the underlying noise is state-dependent (Liu et al., 2021). When $a < 0$, the point $w = 0$ is a local maximum; one hopes that the underlying dynamics escape to infinity, and our goal is to understand the behavior of SGD in this case.

The following proposition and proof show that for the zero-mean and bounded distribution $p(x)$, SGD cannot escape from the local maximum when the learning rate is set to be 1.

**Proposition 1.** *Let $\lambda = 1$ and $a < 0$ and $p(x)$ be the distribution such that $x = 1$ and $x = -1 + a$ with equal probability. Then, $w$ converges to $w = 0$ with probability 1 with the SGD algorithm, independent of the initialization.*

*Proof.* By the definition of the SGD algorithm, we have

$$w_{t+1} = \begin{cases} 0 & \text{with probability } 0.5; \\ (2-a)w_t & \text{otherwise.} \end{cases} \tag{3}$$

Therefore, after $t$ time steps, $w_t$ has at most $2^{-t}$ probability of being non-zero. For infinite $t$, $w_t = 0$ with probability 1. $\square$

This simple toy example illustrates a few important points. First of all, it suggests that one cannot use the expected value of $w$ to study the escape problem of SGD. The expectation value of $w_t$ is

$$\mathbb{E}_x[w_t] = (1 - a/2)\mathbb{E}_x[w_{t-1}] = (1 - a/2)^t w_0 = w_0 e^{\ln(1-a/2)t}, \tag{4}$$

which is an exponentially fast escape from any initialization $w_0$ for $a < 0$. In fact, $\ln(1 - a)$ is exactly the escape time scale of the GD algorithm. This is counter-intuitive: SGD will converge to the local maximum with probability 1, even if its expected value escapes the local maximum exponentially fast.

Alternatively, one might also hope to use the expected loss or the norm $\mathbb{E}_{w_t}[L(w_t)] \sim \|w_t\|_2^2$ as the metric of escape, but it is not hard to see that they suffer the same problem. At time $t$, because $w_t = 0$ with probability $1 - 2^{-t}$ and $w_t = (2 - a)^t w_0$ with probability $2^{-t}$, we have $\mathbb{E}[L(w_t)] \propto w_0^2 e^{2t \ln \frac{2-a}{\sqrt{2}}}$, which also predicts an exponentially fast escape. Again, this fails to reflect the fact that SGD has only a vanishingly small probability of escaping the constructed local maximum. Statistically speaking, the norms of $w_t$ fail to be good metrics to measure the escape rate because these metrics are not robust against outliers. In our example, there is only $2^{-t}$ probability for SGD to escape the local minimum, yet the speed at which this outlier escapes the maximum outweighs the decay in its probability through time, thus contributing more to the expected norm than any other events. This example also shows the difficulty and subtlety of dealing with the escape problem. This example (together with Sec. 5.1) also emphasizes the importance of showing convergence in probability for future works in non-convex optimization because convergence in expectation is not sufficient to guarantee a good empirical performance. The results in the rest of this work can be seen as extensions of this special case to less restrictive settings.

## 5    MAIN RESULTS

In this section, we present our four main theoretical results: (1) SGD may converge to a local maximum; (2) SGD may escape a saddle point arbitrarily slowly; (3) SGD may prefer sharp minima over flat ones; and (4) AMSGrad may converge to a local maximum.

### 5.1    SGD MAY CONVERGE TO A LOCAL MAXIMUM

We show that there are cases where SGD may fail to escape saddle points, while GD can successfully escape; this contradicts the suggestions in Kleinberg et al. (2018) that SGD always escapes a local minimum faster than GD. It is appropriate to begin with the definition of a saddle point.

**Definition 2.** $w$ is said to be a stationary point of $L(w)$ if $\nabla_w L(w) = 0$. $w$ is a minimum of $L$ if, for all $w'$ in a sufficiently small neighborhood of $w$, $L(w') \geq L(w)$. $w$ is said to be a saddle point if $w$ is a stationary point but not a local minimum.

This definition of saddle points, which includes local maxima, agrees with the common definition in the literature (Daneshmand et al., 2018). The following proposition generalizes our warmup example to a family of 1D problems in which SGD converges to a local maximum.

**Proposition 2.** *Let Eq. 2 be the loss function. Let $\lambda > 0$, and $p(x)$ be such that $\mu := \mathbb{E}[\ln|1 - \lambda x|]$ and $s^2 := \mathrm{Var}[\ln|1 - \lambda x|]$; then SGD converges to the local maximum in probability if $\mu < 0$.*

*Proof Sketch.* We first show that $\frac{1}{\sqrt{t}} \ln|w_t/w_0|$ obeys a normal distribution as $t \to \infty$. This allows us to deduce the cumulative distribution function (c.d.f.) of $w$ as $t \to \infty$. The c.d.f. has an bifurcative dependence on the sign of $\mu := \mathbb{E}[\ln|1 - \lambda x|]$. When $\mu < 0$, the distribution of $w_t$ converges to $0$ in probability. See Sec. C.2 for the detailed proof. □

**Corollary 1.** *Let Eq. 2 be the loss function and $p(x) = \frac{1}{2}\delta(x-1) + \frac{1}{2}\delta(x+1-a)$; then SGD converges to the local maximum in probability if*

$$\frac{a}{a-1} < \lambda < \frac{a - \sqrt{a^2 - 8a + 8}}{2(a-1)}. \tag{5}$$

One special feature of this example is that the state-dependent noise vanishes at the saddle point. In this particular example, this is achieved by having an SGD noise proportional to the gradient thus vanishing at the saddle point. This type of structure may be relevant for deep learning because vanishing noise at a stationary point and state-dependent noise can appear at the origin of a deep net, where weights of all layers are zero. Note that the convergence is independent of the actual shape of the distribution $p(x)$, and so may apply to many kinds of distributions besides the Bernoulli distribution we considered. There are three different regimes/phases for this simple escaping problem

(also see Fig. 1-Right). The escape regime $\lambda > \frac{a-\sqrt{a^2-8a+8}}{2(a-1)}$ is also present when $a > 0$; it means that in this regime, SGD will also escape $w_0$ even if it is a local minimum. This regime is not present if we perform a continuous-time analysis (Sec. B), showing that this region is due to the instability of the *discrete-time* SGD algorithm. This kind of escaping is undesirable because, in this regime, the local gradient cannot provide any guidance for minimizing the loss. The region $\frac{a}{a-1} < \lambda < \frac{a-\sqrt{a^2-8a+8}}{2(a-1)}$ is the trapped regime. This is due to the special structure of the minibatch noise – a comparison with the continuous-time analysis suggests that SGD will not be trapped if the noise is not position dependent (Sec. B.1). The small learning rate regime $\lambda < \frac{a}{a-1}$ is the successful escape regime because the SGD noise is of order $\lambda^2$ and becomes negligible at a small $\lambda$ (Liu et al., 2021). This regime corresponds to the lazy learning regime of training: with negligible noise and vanishing gradient, SGD is known to optimize neural networks well (Du et al., 2018). This discussion also justifies our later choice for placing an upper bound to $\lambda$ and focusing only on the second and third regime in Sec. 5.3 and 5.4.

It is important to understand why the previous works that show that the previous guarantees of convergence do not apply to this particular example, for example, Ge et al. (2015). In comparison, the SGD algorithm is not guaranteed to escape the saddle point in this example because two crucial assumptions of Ge et al. (2015) is violated: (1) the gradient does not have a bounded fluctuation around its mean (and so even parameters very far away from the saddle can be brought back close to the saddle point due to the noise), and (2) we do not inject additive noise to the gradient and the actual noise decreases as we approach the saddle point (and so the closer we are to the saddle point, the less likely one can escape). The standard theoretical guarantee of convergence in Ge et al. (2015) thus does not apply to this example. This example shows that the minibatch noise of SGD is very special and can strongly influence convergence. A continuous-time analysis in Section B of this example shows that the convergence to local maxima occurs when the minibatch noise dominates the gradient, namely, when the signal to noise ratio becomes smaller than 1. This suggests the importance of minibatch noise in influencing the dynamics of SGD. This suggests that the signal-to-noise ratio can be an important parameter in SGD dynamics and may deserve more future research.

## 5.2 SGD May Escape Saddle Points Arbitrarily Slowly

In the previous section, we have shown that one can adversarially construct a loss function for a fixed learning rate such that the SGD algorithm converges to a local maximum. This section shows that, even if one can tune the learning rate at will, there is a loss landscape where SGD can take an arbitrarily long time to escape, no matter how one chooses the learning rate. We begin by defining the escape rate.

**Definition 3.** The asymptotic average escape rate of $w$ at learning rate $\lambda$, $\gamma(\lambda)$, is defined as $\gamma := \lim_{t\to\infty} \frac{1}{t} \mathbb{E}_{w_t}[\ln|w_t|]$. The optimal escape rate is defined as $\gamma^* := \sup_\lambda \gamma(\lambda)$.

**Remark.** *This definition can be seen as a generalization of the Lyapunov exponent to a probabilistic setting. We note that this definition of escape rate differentiates this work from the standard literature on escaping saddles, where the focus is on the time scale for reaching a local minimum when saddle points exist (Ge et al., 2015; Daneshmand et al., 2018; Jin et al., 2017), while our definition focuses on the time scale of escaping a specified saddle point. These two time scales can be vastly different.*

As discussed in the previous section, the undesired escape can be achieved by setting the learning rate to be arbitrarily large; however, this is not possible in practice because the learning rate of SGD is inherently associated with the stability of learning, and $\lambda$ usually has to be much smaller than 1.

**Proposition 3.** *Let the loss function be Eq. (2), $a < 0$ and $p(x) \sim \frac{1}{2}\delta(x-1) + \frac{1}{2}\delta(x-a+1)$, where $\delta$ is the Dirac delta function, and $\lambda \le 1$. Then, for any $\epsilon > 0$, there exists $a$ such that $\sup_\lambda \gamma(\lambda) < \epsilon$. Specifically, $\gamma^* = \ln\frac{2-a}{2\sqrt{1-a}}$, and, for any $\epsilon > 0$, $\gamma^* < \epsilon$ if $|a| \le 2\left|-e^\epsilon\sqrt{e^{2\epsilon}-1} - e^{2\epsilon} + 1\right|$.*

Namely, the SGD algorithm can escape the saddle point of Eq. (2) arbitrarily slowly. Intuitively, one expects escaping to be easier as we increase the learning rate (Kleinberg et al., 2018), and that the escape rate should monotonically increase as we increase the learning rate. This example conveys a novel message that the escaping efficiency is not monotonically increasing as the learning rate increases. For this example, the escape rate first decreases and starts to increase only when $\lambda$ is quite large. This example also shows the subtlety in the escaping problem. On the one hand, one needs to avoid a too large learning rate to avoid training instability. On the other hand, one cannot use a too

small learning rate because a small learning rate also makes optimization slow. Thus, there should be a tradeoff between learning speed and learning stability, and we may construct a general theory for finding the best learning rate that achieves the best tradeoff; however, this tradeoff problem is a sufficiently complex problem on its own and is beyond the scope of the present work.

## 5.3 SGD May Prefer Sharper Minima

Modern deep neural networks are often overparametrized and can easily memorize all the training data points. This means that the traditional metrics such as Rademacher complexity cannot be used to guarantee the generalization capability of SGD (Zhang et al., 2017). Nevertheless, neural networks are found to generalize very well. This inspired the hypothesis that SGD, the main training algorithm of neural networks, contains some implicit regularization effect such that it biases the neural network towards simpler solutions (Neyshabur et al., 2017; Soudry et al., 2018). One hypothesized mechanism of this regularization is that SGD selects flat minima over the sharp ones (Hochreiter and Schmidhuber, 1997; Meng et al., 2020; Xie et al., 2021; Liu et al., 2021; Mori et al., 2021; Smith and Le, 2018; Wojtowytsch, 2021b). In this work, we show that this may not be the case because the dynamics and convergence of SGD crucially depend on the underlying mini-batch noise, while the sharpness of the landscape is independent of the noise. Before introducing the results, let us first define the sharpness of a local minimum.

**Definition 4.** Let $w = w^*$ be a local minimum of the loss function $L(w)$; the sharpness $s(w^*)$ of a local minimum $w^*$ is defined as $s(w^*) \coloneqq \mathrm{Tr}[\nabla_w^2 L(w^*)]$. We say that a minimum $w_1^*$ is sharper than $w_2^*$ if and only if $s(w_1^*) > s(w_2^*)$.

In this definition, the sharpness is the trace of the Hessian of the quadratic approximation to the local minimum. Other existing definitions, such as the determinant of the Hessian, are essentially similar to this definition (Dinh et al., 2017).

For an explicit construction, consider the following objective on a 2-dimensional landscape:

$$\hat{L}(w_1, w_2) = \frac{1}{2}\left[-(w_1 - w_2)^2 - (w_1 + w_2)^2 + (w_1 - w_2)^4 + (w_1 + w_2)^4 - 2aw_2^2 + xbw_1^2\right]. \quad (6)$$

for $a, b > 0$ and $p(x) = \frac{1}{2}\delta(x-1) + \frac{1}{2}\delta(x+1)$. The diagonal terms of the Hessian of the loss function is given by

$$\begin{cases} \frac{\partial^2}{\partial w_1^2}\hat{L}(w_1, w_2) = -2 + 12w_1^2 + 12w_2^2; \\ \frac{\partial^2}{\partial w_2^2}\hat{L}(w_1, w_2) = -2 - 2a + 12w_1^2 + 12w_2^2. \end{cases} \quad (7)$$

There are four local minima in total for this loss function. These minima and their sharpnesses are

$$\begin{cases} (w_1, w_2) = (\pm\frac{1}{\sqrt{2}}, 0) & \text{with } s = 4; \\ (w_1, w_2) = (0, \pm\sqrt{\frac{1+a}{2}}) & \text{with } s = 4 + 4a. \end{cases} \quad (8)$$

We see that for positive $a$, the minima at $(0, \pm\sqrt{\frac{1+a}{2}})$ are sharper than the other two minima. We show that SGD will converge to these sharper minima. For this example, we assume that $w_1$ and $w_2$ are bounded because, if initialized sufficiently far from the origin, the fourth-order term will cause divergence.

**Proposition 4.** *Let the loss function be Eq. (6), $|w_1| \leq 1$ and $|w_2| \leq 1$. For any $\lambda < c$ for some constant $c = O(1)$, there exists some $b$ such that if SGD converges in probability, it will converge to the sharper minimum at $(w_1, w_2) = (0, \pm\sqrt{\frac{1+a}{2}})$.*

**Remark.** *In this example, the noise is not full-rank, which is often the case in a deep learning setting for overparametrized networks (Wojtowytsch, 2021a). For example, when weight decay is used, the Hessian of the loss function should be full rank, while the rank of the noise covariance should be proportional to the inherent dimension of the data points, which is in general much smaller than the dimension of the Hessian in deep learning (Ansuini et al., 2019).*

The proof is technical and given in the appendix Sec. C.5. In fact, it has been controversial whether finding a flat minimum can help generalization. For example, Dinh et al. (2017) shows that, for every flat minimum of a ReLU-based net, there exists a minimum that is arbitrarily sharper and generalizes

as well. However, this work does not rule out the possibility that conditioning by using gradient-based optimization, the performance of the sharper minima that gradient descent finds is worse than the performance of the flatter minima. If this assumption is valid, then biasing gradient descent towards flatter minima can indeed help, and the stochasticity of SGD has been hypothesized to help in this regard. However, our construction shows that SGD, on its own, may be incapable of helping SGD converge to a flatter minimum. At least some other assumptions need to be invoked to show that SGD may help. For example, the definition of the neural networks may endow these models with a special kind of Hessian and noise structure that, when combined with SGD, results in a miraculous generalization behavior. However, no previous work has pursued this direction in sufficient depth to our knowledge, and future studies in this direction may be fruitful.

### 5.4 NON-CONVERGENCE OF ADAPTIVE GRADIENTS

Adam (Kingma and Ba, 2014) and its closely related variants such as RMSProp (Tieleman and Hinton, 2012) have been shown to converge to a local maximum even for some simple convex loss landscapes (Reddi et al., 2018a). The proposed fix, named AMSGrad, takes the maximum of all the previous preconditioners in Adam. This section shows that AMSGrad may also converge to a local maximum even in simple non-

$$m_t = \beta_1 m_{t-1} + (1 - \beta_1)\hat{g}_t; \quad (9)$$

$$v_t = \beta_2 v_{t-1} + (1 - \beta_2)\hat{g}_t^2; \quad (10)$$

$$\hat{v}_t = \max(\hat{v}_{t-1}, v_t); \quad (11)$$

$$w_t = w_{t-1} - \frac{\lambda}{\sqrt{\hat{v}_t}} m_t. \quad (12)$$

convex settings. Let $x_t \sim p(x)$ and $\hat{g}_t = \nabla \hat{L}(x_t, w_{t-1})$, the Adam algorithm is given by Eq. (9)-(12), where $v_0 = m_0 = 0$. We have removed the numerical smoothing constant $\epsilon$ from the denominator, which causes no problem if $w_0$ is initialized away from 0. Here, $\beta_1$ is the momentum hyperparameter. $v_t$ is called the preconditioner, and $\beta_2$ is the associated hyperparameter. The standard value for $\beta_1$ is 0.9 and $\beta_2$ is 0.999. In our theory, we only consider the case when $\beta_1 = 0$. The experiment section shows that a similar problem exists when $\beta_1 > 0$ (with additional interesting findings). Intuitively, this is easy to understand because AMSGrad behaves like SGD asymptotically, so we only have to wait for long enough, and the results in previous sections would apply. The construction below follows this intuition.

**Proposition 5.** *Let $w_t \in [-1, 1]$, and $w_0 \neq 0$. For fixed $\lambda < 1$ and the loss function in Eq. (2) there exists $a < 0$ such that the AMSGrad algorithm converges in probability to a local maximum.*

**Remark.** *The proof is given in Sec. C.6. Note that the example we construct is similar to the original construction in Reddi et al. (2018a) that shows that Adam converges to a maximum while AMSGrad succeeds in reaching the global minimum. In their example (with some rescaling), the gradient is a random variable*

$$\hat{g}_t = \begin{cases} 1 & \text{with probability } q \approx 1; \\ c_0 < 0 & \text{with probability } 1 - q \ll 1; \end{cases} \quad (13)$$

*such that the expected gradient is negative, and Adam is shown to converge to the direction opposite to the gradient descent (therefore to a maximum). In our example, the gradient is (roughly)*

$$\hat{g}_t = \begin{cases} w_{t-1} & \text{with probability } 1/2; \\ -(1 + a)w_{t-1} & \text{with probability } 1/2. \end{cases} \quad (14)$$

*Therefore, our example can be seen as a minimal generalization of the Reddi et al. (2018a) example to a non-convex setting.*

## 6 EXPERIMENTAL DEMONSTRATIONS

We perform experiments to illustrate the examples studied in this work. Due to space limitations, we illustrate the escape rate and the convergence to sharper minima example in Appendix Sec. A.

### 6.1 SGD CONVERGES TO A LOCAL MAXIMUM

Numerical results are obtained for the setting described in section 5.1. See Fig. 1-Left. The loss landscape is defined by $L(w) = \frac{1}{4}aw^2$. In this numerical example, we set $\lambda = 0.8$ and $a = -1$, and the histogram is plotted with $2000$ independent runs. We see that the distribution converges to the local maximum at $w = 0$ as the theory predicts. For better qualitative understanding, we also plot

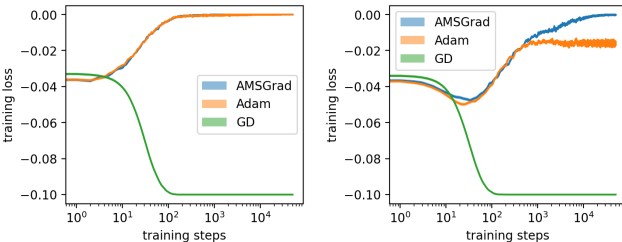

Figure 2: AMSGrad diverges to the local maximum with or without momentum in the example we studied in Sec 5.4. Our result shows that AMSGrad is always attracted to the local maximum, while Adam has the potential of escaping the local maximum with momentum. **Left**: without momentum. **Right**: with momentum.

the empirical phase diagram of this setting in Fig. 1-Right. We perform numerical evaluations for different combinations of $a$ and $\lambda$ on a $a - \lambda$ plane, and for each combination, we plot the percentage of $w$ that escapes the central bin (white is $100\%$ and **dark blue** is $0\%$). For completeness, we also plot the case when $a > 0$, i.e., when the critical point at $w = 0$ is a local minimum. The orange line shows where the phase transition is expected to happen according to Eq. (5). The fact that the orange line agrees exactly with the empirical phase transition line confirms our theory.

We also numerically study a closely related fourth-order loss landscape, defined as $\hat{L}(w) = \frac{1}{2}xw^2 + \frac{1}{4}w^4$, where the distribution $p(x)$ is the same as before. The expected loss $L(w)$ has two local minima located as $w = \pm\sqrt{-2a}$. We are interested in whether SGD can converge to these two points successfully; note that, without the fourth-order term, this loss is the same as the quadratic loss we studied. See Fig. 1-Middle. As before, $a = -1$ and $\lambda = 0.8$. We see that the distribution of $w$ also concentrates towards the local maximum after training, and no $w$ is found to converge to the two local minimum even if a significant proportion of $w$ is initialized close to these two minima. A phase diagram analysis is given in the Sec. A.1.

## 6.2 NON-CONVERGENCE OF AMSGRAD

In this section, we illustrate the example of the convergence behavior of AMSGrad described in Sec. 5.4; for reference, we also plot the behavior of Adam and gradient descent for this example. See Fig. 2. The experiment is the average over 2000 runs, each with $5 \times 10^4$ steps. The uncertainty is reflected by the shaded region (almost invisibly small). The loss function is the same as discussed in Sec. 5.4 with $a = -0.1$. For illustration purposes, we set $\lambda = 0.2$ and $\beta_2 = 0.999$ for both Adam and AMSGrad. When momentum is used, we set $\beta_1 = 0.9$. GD is run with a learning rate of $0.01$. Without momentum, both Adam and AMSGrad converge to the local maximum with almost the same speed. When momentum is added, AMSGrad still converges to the local maximum. Curiously, Adam is no longer attracted to the local maximum but also remains away from the local minimum. This suggests that Adam, with momentum, might have a better capability of escaping saddle points than AMSGrad and might be preferable to AMSGrad in a non-convex setting.

## 6.3 A TOY NEURAL NETWORK EXAMPLE

We have now illustrated several SGD phenomena we studied in artificial settings, inspiring the natural question of whether they occur for an actual neural network. Here we construct a neural-net-like optimization problem to show that the problems with convergence might indeed arise for a neural network. The toy neural network function is given by $f(x) = w_2\sigma(w_1 x)$, where $w_1, w_2 \in \mathbb{R}$. $\sigma$ is the nonlinearity, and we let $\sigma(x) = x^2$ as a minimal example. This is the simplest kind of nonlinear feedforward network one can construct (2 layer with a single hidden neuron). We also pick a minimal dataset with a single data point: $x = 1$ with probability 1 and $y \in \{-1, 2\}$, each with probability 0.5, i.e., the label has a degree of inherent uncertainty, which is often the case in real problems. We use mean squared error (MSE) as the loss function. Therefore, the expected loss is $L(w_1, w_2) = \frac{1}{2}(w_2 w_1^2 + 1)^2 + \frac{1}{2}(w_2 w_1^2 - 2)^2$. The global minimum $L^* = 2.25$ is degenerate, and is achieved when $w_1 = 1/\sqrt{2w_2}$. There is also a manifold of saddle points given by $\{(w_1, w_2)|w_1 = 0, w_2 \geq 0\}$, all with $L = 2.5$. Despite the simplicity of this example, it contains a few realistic features in a realistic problem, including (1) inherent uncertainty in the data, (2) a nonlinear hidden layer, and (3) a degenerate minimum with zero eigenvalues in the Hessian.

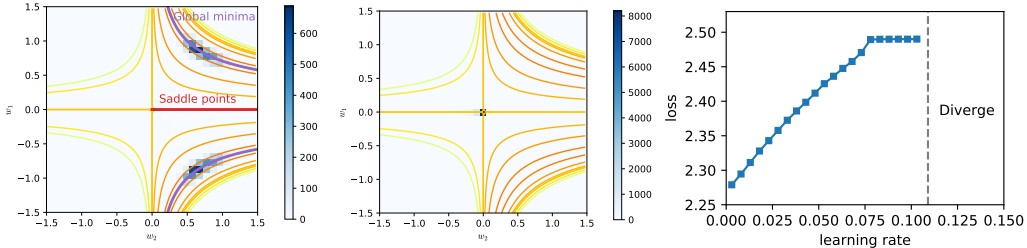

Figure 3: Convergence of a two-layer one-neuron neural network to a saddle point. The blue region shows the empirical density of converged parameter distribution. **Left**: $\lambda = 0.001$ at step 10000 converges to global minima. **Mid**: $\lambda = 0.1$ at step 10000 converges to a saddle point. **Right**: Average loss in equilibrium as a function of learning rate. The loss function diverges for learning rates larger than 0.108.

More importantly, most previous works on the convergence or escaping behavior of SGD are inapplicable to this example. One dominant assumption is the $\rho$-Hessian Lipschitz property (Jin et al., 2017; Ge et al., 2015), which does not hold for this example in particular. Another common assumption is the PL condition (Karimi et al., 2020; Wojtowytsch, 2021a; Vaswani et al., 2019), which does not apply due to the existence of the saddle point. One recent assumption is that the loss function is 1-point convex for all points (Kleinberg et al., 2018), which is also ruled out because $(0, 0)$ is not 1-point strongly convex. Another relevant assumption is the correlated negative curvature assumption, which also does not hold for $(0, 0)$. We explain in Sec. A.5 in detail why these conditions are violated.

See Fig. 3 for the experimental results with this example. Here, $w_1$ is initialized uniformly in $[-1, 1]$; $w_2$ is initialized uniformly in $[0, 1]$ to be closer to the global minimum and away from the saddle point (standard initialization such as initializing in $[-1, 1]$ does not change the conclusion). The left figure shows the stationary distribution of the model parameter at a small learning rate ($\lambda = 0.001$). All the parameters are located in the global-minimum valley as expected. In contrast, when the learning rate is large ($\lambda = 0.1$), the central figure shows that all models (1000 independent runs) converge to the saddle point at $(0, 0)$. The right figure investigates this change more systematically and shows the change in the average stationary training loss of the models as we increase the learning rate from 0.001 to 0.15. We see that for a small learning rate, the training loss is close to that of the global minima ($L = 2.25$), and for a significant range of large learning rates, the model invariably converges to a saddle point ($L = 2.5$). One additional interesting observation is that the model diverges at $\lambda \approx 0.11$, and there is almost no sign of such divergence (such as increased fluctuation) when the learning rate is close to the divergent threshold. This example shows the relevance of our results to the study of neural networks. In fact, it has often been noticed that at convergence, many large-scale neural networks in real tasks exhibit negative eigenvalues in the Hessian (Alain et al., 2019; Granziol et al., 2019). The existence of negative eigenvalues at a late time suggests either the possibility that the algorithm has yet to escape or that it has indeed been attracted to such saddle points; if the latter is true, then our work offers an explanation. The emergence of negative eigenvalues at the end of training serves as indirect evidence that our result may be relevant for larger neural networks. It thus remains an important open question to prove (and identify the condition of) or disprove the convergence of larger and deeper neural networks to a local maximum.

## 7 DISCUSSION

We have shown that when the learning rate is not carefully chosen or scheduled, SGD can exhibit many undesirable behaviors, such as convergence to local maxima or saddle points. The limitation of our work is clear. At best, all the constructions we made are minimal and simplistic toy examples that are far from practice, and investigating whether the discovered messages are relevant for deep learning or not is the one important immediate future step. Other relevant questions include: (1) Does convergence to saddle points help or hurt generalization? (2) If it hurts, how can we modify SGD to avoid saddle points better? We suspect changing learning rates, changing batch size, or injecting noise may help, but a convincing theoretical guarantee in a realistic setting is yet lacking. (3) If saddle points do not have worse generalization, our results motivate for understanding why. This is especially relevant because we showed that using a large learning rate is more likely to converge to saddle points, while previous works have shown that using a large learning rate can improve generalization; combined, this may imply that certain saddle points may have intriguing but unknown regularization effects.

ACKNOWLEDGMENT

We thank the anonymous reviewers for providing detailed and constructive feedback for our draft. Ziyin thanks Jie Zhang for the help during the writing of this manuscript. Ziyin is financially supported by the GSS Scholarship from the University of Tokyo. BL acknowledges CNRS for financial support and Werner Krauth for all kinds of help. JS gratefully acknowledges support from the National Science Foundation Graduate Fellow Research Program (NSF-GRFP) under grant DGE 1752814. This work was supported by KAKENHI Grant Numbers JP18H01145 and JP21H05185 from the Japan Society for the Promotion of Science.

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

## A    ADDITIONAL EXPERIMENTS

### A.1    PHASE DIAGRAM OF THE FOURTH-ORDER LOSS FUNCTION

With two proper local minima, the fourth-order loss function is a more realistic loss function than the one we considered in Sec. 5.1. We also performed one experiment in the main text (See Fig. 1). Here, we plot its empirical phase diagram in Figure 4. We see that for this loss landscape also, there is some region such that for all $\lambda$, SGD converges to the local maximum. This loss landscape is very difficult to study in discrete time as it is known to lead to chaotic behavior at large learning rate (May, 1976). Therefore, we alternatively try to understand this landscape using continuous approximation. See Sec. B.

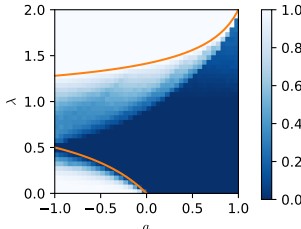

Figure 4: Escape probability from the local maximum as a function of $a$ and $\lambda$ with fourth-order loss landscape. The parameters space is divided into an absorbing phase where $w$ is attracted to the local maximum and an active phase where $w$ escapes to infinity successfully. The orange line is the theoretical phase transition line for the quadratic loss function. We see that when $\lambda$ is small, the line based on quadratic loss also gives good agreement with the fourth-order loss. This suggests that part of this result is universal and independent of the details of the loss function.

### A.2    ESTIMATING THE ESCAPE RATE

In the escape rate experiments, the empirical results are obtained from the proposed approximation. Here, we show that it is valid. See Fig. 5, where we plot the estimated $\gamma$ as a function of the training step. We see that the estimated value converges within about 20 steps. We, therefore, estimate the escape rates at time step 100 in the main text.

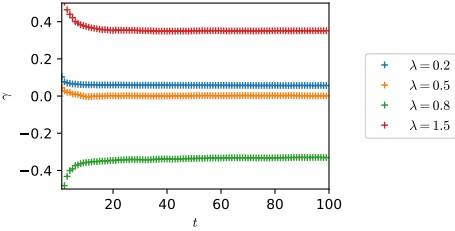

Figure 5: Escape rate $\gamma$ as a function of time step $t$ obtained when $a = -1$, showing that $\gamma$, defined in Equation (73), is indeed a well-defined quantity when $t$ is large. $\gamma$ becomes stable after roughly 40 steps.

### A.3    ESCAPE RATE EXPERIMENTS

This section illustrates the slow escape problem studied in Sec. 5.2. See Fig. 6. $\gamma$ is calculated by averaging the first 50 time steps across 2000 independent runs. We see that, as our theory predicts, as $|a|$ gets closer to 0, the optimal escape rate decreases towards 0. This implies that there exists a landscape such that SGD is arbitrarily slow at learning, independent of parameter tuning. One additional observation is that the optimal learning rate is neither too large nor too small, i.e., there seems to be a tradeoff between the escape speed and escape probability (and between the speed and stability). See also the discussion at the end of Sec. 5.2.

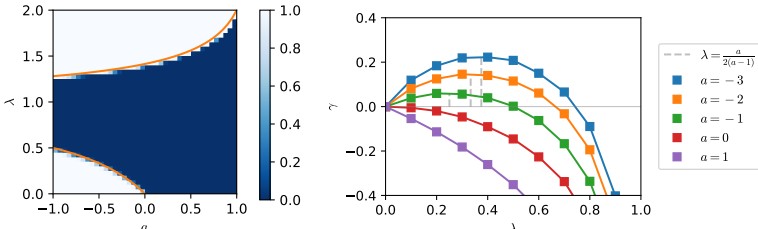

Figure 6: Escape rate $\gamma$ as a function of learning rate $\lambda$ with quadratic loss landscape, obtained by simulations. We note that the escape-rate analysis yields results which are compatible with the phase diagram.

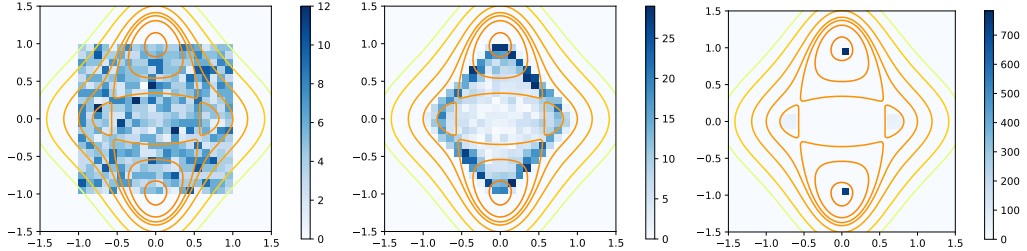

Figure 7: Evolution of the distribution of $w$ in a landscape where two flat minima and two sharp minima exist. As Proposition 4 shows, the model parameters converge to the sharper minima even if initialized in the flat minimum. **Left**: Initialization. **Mid**: step 2. **Right**: step 10000.

### A.4 CONVERGENCE TO THE SHARPER MINIMUM

We use the same 2-dimensional loss landscape defined in Sec. 5.3. The experiment is run with 2000 independent simulations and learning rate $\lambda = 0.05$. See Fig. 7, where we overlap the underlying landscape with the empirical distribution (the heat map). We see that, even though the initialization overlaps significantly with the local flatter minimum, all points converge to the sharper minimum, as the theorem predicts.

### A.5 INADEQUACY OF STANDARD ASSUMPTIONS FOR THE NEURAL NETWORK EXAMPLE

In this section, we show that the standard assumptions in the previous literature do not hold for the example studied in Sec. 6.3. One common assumption is that the loss function is $\rho$-Hessian Lifschitz.

**Definition 5.** A loss function $L$ is said to be $\rho$-Hessian Lifschitz if, for some $\rho > 0$,

$$\forall \mathbf{w}, \mathbf{w}', \|\nabla^2 L(\mathbf{w}) - \nabla^2 L(\mathbf{w})\| \leq \rho \|\mathbf{w} - \mathbf{w}'\|. \tag{15}$$

This certainly does not hold for the example we considered. Explicitly, let $w_2 = 0$, the Hessian $H(\mathbf{w})$ of the loss function is

$$H_{w_1,w_1} = H_{w_1,w_2} = H_{w_2,w_1} = 0, \tag{16}$$

and

$$H_{w_2,w_2} = -w_1^2. \tag{17}$$

Let $w_1 \to \infty$ violates the $\rho$-Hessian Lifschitz property.

Another common assumption for the non-convex setting is the PL condition.

**Definition 6.** Let $L^*$ be the value of $L$ at the global minimum. A loss function $L$ is said to satisfy the PL (Lojasiewitz) condition if

$$\forall \mathbf{w}, \|\nabla L(\mathbf{w})\|^2 \geq \mu(L(\mathbf{w}) - L^*), \tag{18}$$

for some $\mu > 0$.

This also does not hold due to the existence of the saddle point. Let $(w_1, w_2) = (0, 0)$. The gradient is zero, but the right-hand side is greater than 0.

Recently, the correlated negative curvature assumption has been proposed to study the escaping behavior of SGD.

**Definition 7.** Let $\mathbf{v_w}$ be the eigenvector of the minimum eigenvalue of the Hessian $H(\mathbf{w})$. $L$ satisfies the correlated negative curvature assumption if, for some $\gamma > 0$,

$$\forall \mathbf{w}, \mathbb{E}_x[\langle \mathbf{v_w}, \nabla L(x, \mathbf{w})\rangle^2] > \gamma \tag{19}$$

where $x$ is the data point.

This also does not hold. The point $(0,0)$ violates this condition because $\nabla L(x, \mathbf{w}) = 0$ for all $x$ at $(w_1, w_2) = (0,0)$. In fact, it is exactly this point that violates this condition that the SGD converges to in this example.

Recently, the one point strongly convex assumption has been proposed to study the escaping behavior of SGD.

**Definition 8.** A loss function is said to satisfy the $c$-one point strongly convex condition with respect to $\mathbf{w}^*$ for all gradient noise $\epsilon$, $c > 0$, define $\mathbf{v} = \mathbf{w} - \lambda \nabla L(\mathbf{w})$, we have

$$\langle -\nabla \mathbb{E}_\epsilon L(\mathbf{v} - \lambda \mathbf{w}, \mathbf{w}^* - y) \geq c\|\mathbf{w}^* - \mathbf{v}\|_2^2, \tag{20}$$

where $\epsilon$ is the random noise caused by minibatch sampling.

This assumption is equivalent to that the loss landscape is strongly convex after convoluting with the noise. This condition implies that there is only a single stationary point. However, this is not the case, consider any point with $w_1 = 0$ and $w_2 \leq 0$. These points have zero gradient for $w_1 \leq 0$ and so $\epsilon = 0$ with probability $1$. Then, these points all have the same loss after the convolution:

$$\mathbb{E}[L(\mathbf{v} - \lambda \epsilon)] = L(\mathbf{v}), \tag{21}$$

which either imply that the landscape is not convex or that there is more than $1$ stationary point, and so the system is not one point strongly convex [1].

---

[1] One notices that the origin, where all the parameters are zero, is a special point in the problem. In fact, the origin may be a very special point in the landscape of deep neural networks in general. For example, see Ziyin et al. (2022a).

## B   CONTINUOUS-TIME APPROXIMATION WITH FOKKER-PLANCK EQUATION

In this section, we study the examples we studied in the main text in a continuous-time approximation, in order to understand the unique aspects of discrete-time SGD. Compared with the discrete-time analysis, continuous-time analysis is usually more powerful in terms of calculation: it is able to deal with more complicated potentials.

### B.1   FOKKER-PLANCK EQUATION AND ITS STATIONARY DISTRIBUTION

Recall that the SGD update takes the form (see Sec. 2)

$$\Delta w_t = -\lambda \nabla L + \lambda \sqrt{C} \eta_t, \tag{22}$$

when $\lambda < 1$, the above equation may be approximated by a continuous-time Ornstein-Uhlenbeck process (Mandt et al., 2017)

$$dw(t) = -\lambda \nabla L dt + \frac{\lambda}{\sqrt{S}} \sqrt{C(w)} dW(t), \tag{23}$$

where $\lambda$ is the learning rate; we have also introduced $S$ as the batch size. $dW(t)$ is a stochastic process satisfying

$$\begin{cases} dW(t) \sim \mathcal{N}(0, dtI), \\ \mathbb{E}[dW(t)dW(t')] \propto \delta(t - t'). \end{cases} \tag{24}$$

By the definition of the underlying discrete-time process, the term $dw(t)$ depends only on $w(t)$ and has no dependence on $w(t + dt)$. Thus, the stochastic integration should be interpreted as Ito.

For simplicity, we only consider one-dimensional version of the Fokker-Planck, which is

$$\frac{\partial P[w, t|w(0), 0]}{\partial t} = -\frac{\partial}{\partial w} J(w, t|w(0), 0), \tag{25}$$

where $J[w, t|w(0), 0]$ is the probability flow. The current $J[w, t|w(0), 0]$ is:

$$J[w, t|w(0), 0] = \lambda \frac{\partial L}{\partial w} P[w, t|w(0), 0] + \frac{\lambda^2}{2S} \frac{\partial}{\partial w} \{C(w)P[w, t|w(0), 0]\}. \tag{26}$$

Assuming that the SGD dynamics is ergodic, there is a unique stationary distribution for $w$ when $t \to \infty$ that can be found to be

$$P(w) \propto \frac{1}{C(w)} \exp\left[-\frac{2S}{\lambda} \int dw \frac{1}{C(w)} \frac{\partial L}{\partial w}\right]. \tag{27}$$

We will apply this equation to study the relevant problems in this work.

### B.2   A GENERAL CASE

In this section, we consider (a slightly more general version of) the fourth-order potential we studied in the main text. The average loss landscape in this model is

$$L(w) = \frac{1}{4} aw^2 + \frac{1}{4} bw^4, \tag{28}$$

where $b > 0$, guarantees that the $w$ is bounded regardless the value of $a$. In the limit of $b \to 0$, this function reduces to saddle point problem we studied in Sec. 5.1. Besides the SGD noise, additive noise is also present in the dynamics. The variance of the additive noise has no dependence on $w$. When the additive noise is present, the update rule (equation of motion) is

$$dw(t) = -[\lambda(a/2)w(t) + bw^3(t)]dt + \lambda \eta_m(t) + \lambda \eta_a(t), \tag{29}$$

where both $\eta_m(t)$ and $\eta_a(t)$ are both Ornstein-Uhlenbeck process, denoting multiplicative noise and additive noise respectively. The $w$-dependent covariance of $w$ is $C(w) = w^2$. The SGD noise is thus a multiplicative noise. $\eta_m(t)$ and has a variance $\frac{w(t)^2}{S} dt$, while the additive noise $\eta_a(t)$ has a variance of $\sigma^2 dt$, where $\sigma$ is a positive constant denoting the strength of the additive noise. If the

multiplicative noise is seen as a part of the loss landscape, (2) coincides with the 2nd-order term in the loss landscape. There is no correlation between the additive noise and the SGD noise, i.e. $\mathbb{E}[\eta_m(t)\eta_a(t')] = 0$. The additive noise can be seen as the artificially injected noise, a technique sometimes used for aiding the escape or for Bayesian learning purposes (Jin et al., 2017; Welling and Teh, 2011).

The additive noise vanishes in the limit of $\sigma \to 0$, and the model becomes a normal SGD with 4th-order loss function. It is always possible to define a noise $\eta'(t)$, whose contribution is equivalent to the contribution of both the SGD and the additive noise, and the equation of motion becomes

$$dw(t) = -[\lambda(a/2)w(t) + bw^3(t)]dt + \lambda\eta'(t). \tag{30}$$

This transformed noise $\eta'(t)$ thus has 0 mean and a variance of $\left(\frac{w(t)^2}{S} + \sigma^2\right)dt$. Define $\eta(t) = \eta'(t)/\sqrt{\frac{w(t)^2}{S} + \sigma^2}$, the equation of motion becomes

$$dw(t) = -\lambda[(a/2)w(t) + bw^3(t)]dt + \lambda\sqrt{\frac{w(t)^2}{S} + \sigma^2}\eta(t). \tag{31}$$

Comparing with (23), one finds

$$C(w) = w^2 + S\sigma^2. \tag{32}$$

The solution of the corresponding Fokker-Planck equation is

$$
\begin{aligned}
P(w) &\propto \frac{1}{w^2 + S\sigma^2}\exp\left[-\frac{2S}{\lambda}\int\frac{(a/2)w + bw^3}{w^2 + S\sigma^2}dw\right]\\
&= (w^2 + S\sigma^2)^{-1}\exp\left[-\frac{2Sb}{\lambda}\int\frac{\left(\frac{a}{2b} - S\sigma^2\right)w + w^3 + S\sigma^2 w}{w^2 + S\sigma^2}dw\right]\\
&= (w^2 + S\sigma^2)^{-1}\exp\left[-\frac{2Sb}{\lambda}\int wdw - \frac{2Sb}{\lambda}\left(\frac{a}{2b} - S\sigma^2\right)\int\frac{w}{w^2 + S\sigma^2}dw\right]\\
&= (w^2 + S\sigma^2)^{-1}\exp\left[-\frac{Sb}{\lambda}w^2 - \left(\frac{Sa}{2\lambda} - \frac{S^2\sigma^2 b}{\lambda}\right)\ln\left(w^2 + S\sigma^2\right)\right].
\end{aligned}
$$

Further simplification yields

$$P(w) \propto (w^2 + S\sigma^2)^{-1-\frac{Sa}{2\lambda} + \frac{S^2 b\sigma^2}{\lambda}}\exp\left[-\frac{Sb}{\lambda}w^2\right]. \tag{33}$$

The function $P(w)$ is finite everywhere and decays exponentially to 0 when $w \to \infty$, regardless of the values of the parameters $a, b, \sigma, S$. This indicates that $\int_{-\infty}^{\infty} dw(w^2 + S\sigma^2)^{-1-\frac{Sa}{2\lambda} + \frac{S^2 b\sigma^2}{\lambda}}\exp\left[-\frac{Sb}{\lambda}w^2\right]$ has a well-defined value. As a consequence, there is no concentration of measure. Thus, $w(t)$ can be fall into any interval with finite probability after big enough $t$, indicating that $w(t)$ can be arbitrarily far away from 0. In practice, this means that $w(t)$ escapes from the saddle point at $w = 0$ regardless of the value of the parameters.

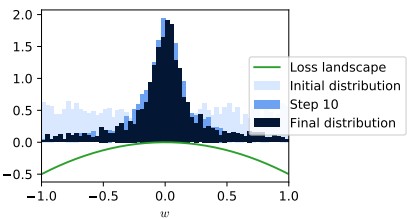

Figure 8: Stationary distribution of $w$ with additive noise of $\sigma = 0.1$ with quadratic loss landscape. All the other parameters are identical to those in Fig.1. This distribution has finite width.

The effect of additive noise is better understood when $b = 0$. In this case, which corresponds to the case shown in Fig.1, $w$ neither concentrates at 0, nor escapes to $\pm\infty$. Instead, it stays near 0 without

converging to it. The stationary distribution of $w$ when $b = 0$, obtained by numerical simulation, is shown in Fig.8. The other settings are identical to the ones in Fig.1. The stationary distribution exists, but $w$ does not concentrate at $w = 0$ in the stationary distribution. First of all, this figure shows that the additive noise helps SGD to escape saddle point. If the designed landscape is a part of a realistic landscape, having a broader distribution means having more chance of being attracted by another minimum. However, this figure also shows that $w$ would stay in the neighborhood of $w = 0$ until the noise is big enough. Thus, having additive noise and long enough training time does not guarantee the efficiency of escape.

In the language of Bayesian inference, $\ln[P(w)]$ is the likelihood of $w$. $\hat{w}$, the most probable value of $w$, is defined by the relation

$$\left.\frac{d\ln[P(w)]}{dw}\right|_{w=\hat{w}} = 0. \tag{34}$$

which reads

$$
\begin{aligned}
0 &= \left\{\frac{d}{dw}\left[\left(-1 - \frac{Sa}{2\lambda} + \frac{S^2 b\sigma^2}{\lambda}\right)\ln\left[(w^2 + S\sigma^2)\right] - \frac{Sb}{\lambda}w^2\right]\right\}\Bigg|_{w=\hat{w}} \\
&= \left\{2w\frac{-1 - \frac{Sa}{2\lambda} + \frac{S^2 b\sigma^2}{\lambda}}{w^2 + S\sigma^2} - \frac{2Sb}{\lambda}w\right\}\Bigg|_{w=\hat{w}} \\
&= \left\{w\left[-1 - \frac{Sa}{2\lambda} + \frac{S^2 b\sigma^2}{\lambda} - \frac{Sb}{\lambda}(w^2 + S\sigma^2)\right]\right\}\Bigg|_{w=\hat{w}} \\
&= \hat{w}\left(\frac{\lambda}{Sb} + \frac{a}{2b} + \hat{w}^2\right).
\end{aligned}
$$

The likelihood maximizer is

$$\hat{w} = \begin{cases} 0, & a > -\frac{2\lambda}{S}; \\ \pm\sqrt{-\frac{\lambda}{Sb} - \frac{a}{2b}}, & a < -\frac{2\lambda}{S}. \end{cases} \tag{35}$$

When $a > -\frac{2\lambda}{S}$, the maximum likelihood parameter is $w = 0$. When $a < -\frac{2\lambda}{S}$, $w = \pm\sqrt{-\frac{\lambda}{Sb} - \frac{a}{2b}}$ equally likely. This resembles the phase transition in statistical physics and we call $-\frac{2\lambda}{S}$ the critical value of $a$, or, the "critical $a$". For the energy landscape, when $a > 0$ there is only one global minimum at $w = 0$, and when $a < 0$ there are two global minima at $w = \pm\sqrt{-\frac{a}{2b}}$. Comparing with the maximum likelihood solutions, we see that the likelihood maximizer given by SGD is in fact a biased estimator of the underlying minima.

There is a bias term introduced by the SGD noise in both the critical $a$ and the value of the most probable $w$. This term vanishes when $S \to \infty$, i.e. when the SGD noise vanishes. This indicates that, when the noise is state-dependent, there is no reason to expect SGD to be an unbiased or consistent estimator of the minimizer, as some works assume.

### B.3 4-TH ORDER POTENTIAL WITH MULTIPLICATIVE NOISE

In this subsection, we study a 4-th order loss landscape in SGD dynamics without additive noise. The average loss function with 4th-order term is

$$L(w) = \frac{aw^2}{4} + \frac{bw^4}{4}; \tag{36}$$

the SGD update rule (i.e., the equation of motion) in this case is

$$dw(t) = -\lambda[(a/2)w(t) + bw^3(t)]dt + \lambda\frac{1}{\sqrt{S}}\eta(t)w(t), \tag{37}$$

where the definition of $\eta$ is identical to the previous example. The solution of corresponding stationary Fokker-Planck equation is

$$P(w) \propto w^{-2-\frac{Sa}{\lambda}}\exp\left[-\frac{Sb}{\lambda}w^2\right]. \tag{38}$$

When

$$- \frac{Sa}{2\lambda} < 1, \tag{39}$$

$P(w)$ diverges at $w = 0$. The normalization factor, i.e. integral $\int_{-\infty}^{+\infty} dw w^{-2-\frac{Sa}{\lambda}} \exp\left[-\frac{Sb}{\lambda} w^2\right]$, also diverges due to the divergence at $w = 0$. This solution could be seen as a limit of (33) when the additive noise vanishes. As the strength of the additive noise $\sigma \to 0$, the normalization factor keep growing while the distribution remains normalized. For $w \neq 0$, the function $w^{-2-\frac{Sa}{\lambda}} \exp\left[-\frac{Sb}{\lambda} w^2\right]$ is always finite. Thus, in the limit $\sigma \to 0$, $P(w)$ approaches $0$ everywhere except for at $w = 0$. It is straight forward to check that the normalization factor diverges slower that $P(0)$ as $\sigma \to 0$. As a consequence, when $\sigma = 0$, $P(w)$ diverges at $w = 0$ and $P(w)$ becomes a Dirac delta function. This corroborates with the result in Sec. 5.3 that a small $a$ leads to a concentration of measure towards the local maximum.

Compared with (33), one finds that the additive noise prevents the concentration of measure. Thus, in practice, the additive noise helps SGD escape the local minimum or saddle point. However, the existence of additive noise does not change the critical $a$ and the position of the peaks.

### B.4 QUADRATIC POTENTIAL

At last, we consider the continuous approximation of the model treated in the Sec. 5.1. Let the loss function be

$$\hat{L} = \frac{x}{2} w^2, \tag{40}$$

and we have

$$\begin{cases} \mathbb{E}_x[\hat{g}_t] = \frac{a}{2} w_t \\ C(w_t) = \frac{1}{S} \mathbb{E}\left[\nabla \ell \nabla \ell^{\mathrm{T}}\right] - \frac{1}{S} \nabla; \quad L(\mathbf{w}_t) \nabla L(\mathbf{w}_t)^{\mathrm{T}} = \frac{1}{S} \mathbb{E}[x^2] w_t^2. \end{cases} \tag{41}$$

The SGD update rule (i.e., the equation of motion) in 1d is

$$\Delta w(t) = -\lambda \frac{\partial}{\partial w} \hat{L} \tag{42}$$

$$= -\lambda x w(t). \tag{43}$$

With the continuous approximation, the equation of motion becomes

$$dw(t) = -\lambda(a/2) w(t) dt + \lambda \frac{1}{\sqrt{S}} w(t) \eta(t), \tag{44}$$

where $\eta(t) \sim \mathcal{N}(0, \sqrt{dt})$. By comparing with the 1d Fokker-Planck equation,

$$\begin{cases} L(w) = \frac{a}{4} w^2; \\ B(w) = w. \end{cases} \tag{45}$$

The stationary distribution of $w$ is

$$P(w) \propto \frac{1}{w^2} \exp\left[-\frac{S}{\lambda} \int dw \frac{a}{w}\right] \propto w^{-2-\frac{Sa}{\lambda}}. \tag{46}$$

The only difference between the solution in this case and the one in the 4th-order-loss case is the exponential factor, showing that the 4th-order potential does nothing but keeping $w(t)$ bounded. The function $w^{-2-\frac{Sa}{\lambda}}$ diverges at $w = 0$ or $w = \pm\infty$ depending on the value of $a$. However, the stationary distribution, being a limit of (33), is well defined. In the case that it diverges at $w = 0$, $P(w)$ becomes a delta function. On the contrary, there are two peaks infinitely far away from $0$ when $w^{-2-\frac{Sa}{\lambda}}$ diverges at $w = \pm\infty$. Being able to escape requires that the probability measure of $w$ does not concentrate at $0$, corresponding to

$$- \frac{Sa}{2\lambda} > 1. \tag{47}$$

This straight line agrees approximately with the boundary of the lower branch of the phase diagram in Fig. 9. This condition of escaping is worth interpretation. Note that $a$ is the local curvature and reflects the strength of the gradient signal. In contrast, the strength of the SGD noise is proportional

to $\lambda/S$.[2] The term $Sa/2\lambda$ is thus the signal to noise ratio of this learning problem, and the escaping condition is exactly when the signal becomes larger than the noise. This analysis, therefore, pinpoints the cause of the convergence to saddle points to be the dominance of the SGD noise over the gradient.

Also, this example raises an interesting question regarding the mechanism of SGD that causes a convergence to the local maximum. From the perspective of the types of convergence, the SGD mechanism behind proposition 1 may be different from that of proposition 2. However, from the perspective of continuous-time analysis, there is really the same mechanism that is governing the SGD dynamics behind proposition 1 and 2 (namely, the fact that the noise has dominated the gradient).

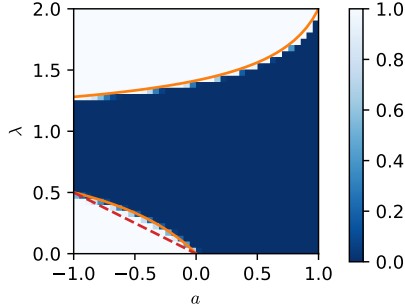

Figure 9: Escape probability as a function of $a$ and $\lambda$. The parameters space is divided into an absorbing phase where $w$ is attracted to the local maximum (in **dark blue**) and an active phase where $w$ successfully escapes the two central bins (in white). The orange line denotes the analytical convergence bound on $\lambda$ as a function of $a$ obtained in the discrete-time calculation, while the red dashed line denotes the same bound obtained by the continuous-time calculation. Here the batch size $S$ is set to 1. The result given by the continuous-time process agrees well with the discrete-time process in the small $\lambda$ small $a$ limit.

---

[2]See Ziyin et al. (2022b), for example.

## C  DELAYED PROOFS

### C.1  TWO FREQUENTLY USED LEMMAS

We first prove the following lemma regarding the limiting distribution of $\ln|w_t|$.

**Lemma 1.** *Let the loss function be $\hat{L}(w) = \frac{1}{2}xw^2$, $x \sim p(x)$ such that $\mathrm{Var}[x] = \sigma^2$ and $\mathbb{E}_x[x] = a < 0$ and that $p(x)$ is continuous in a $\delta$-neighborhood of $x = 1$, and $w_0 \neq 0$. Then, for $w_t$ generated by SGD after $t$ time steps,*

$$\frac{1}{\sqrt{t}}(\ln|w_t/w_0| - \mu) \to_d \mathcal{N}(0, s^2) \tag{48}$$

*where $\mu = \mathbb{E}_x[\ln|1 - \lambda x|]$ and $s^2 = \mathrm{Var}[\ln|1 - \lambda x|]$, where $\lambda > 0$ is the learning rate.*

*Proof.* After $t$ steps of SGD, we have

$$w_t = \prod_{i=1}^{t}[1 - \lambda x_i]w_0. \tag{49}$$

This leads to

$$\ln\left|\frac{w_t}{w_0}\right| = \sum_{i=1}^{t}\ln|1 - \lambda x_i|. \tag{50}$$

Now, since $p(1-x)$ is continuous in the neighborhood of 1 by assumption, we can apply Proposition 6 (appendix) to show that the first and second moment of $\ln|1 - \lambda x|$ exists, whereby we can apply the central limit theorem to obtain

$$\frac{1}{\sqrt{t}}\left(\ln\left|\frac{w_t}{w_0}\right| - t\mathbb{E}[\ln|1 - \lambda x|]\right) \to_d \mathcal{N}(0, s^2), \tag{51}$$

where $s^2 = \mathrm{Var}[\ln|1 - \lambda x|]$. This proves the lemma. □

**Proposition 6.** $\mathbb{E}_{x\sim p(x)}[\ln^2|x|]$ *is finite if the second moment of $p(x)$ exists and if $p(x)$ is continuous in the neighborhood of $x = 0$.*

*Proof.* Since $p(x)$ is continuous in the neighborhood of $x = 0$, then there exists $\delta > 0$ such that for all $|\epsilon| < \delta$, $|p(\delta) - p(x)| \leq c$ for some $c > 0$. This means that we can divide the integral over $p(x)$ into three regions:

$$\mathbb{E}_{x\sim p(x)}[\ln|x|^2] = \int_{-\infty}^{\infty} dx p(x)\ln^2|x| \tag{52}$$

$$= \int_{-\infty}^{-\delta} dx p(x)\ln^2|x| + \int_{-\delta}^{\delta} dx p(x)\ln^2|x| + \int_{\delta}^{\infty} dx p(x)\ln^2|x|. \tag{53}$$

The second term can be bounded as

$$\int_{-\delta}^{\delta} dx p(x)\ln^2|x| \leq [p(0) + c]\int_{-\delta}^{\delta} dx \ln^2|x| \tag{54}$$

$$= 2[p(0) + c]\int_{0}^{\delta} dx \ln^2 x \tag{55}$$

$$= 2[p(0) + c]x(\ln^2 x - 2\ln x + 2)\Big|_{0}^{\delta} \tag{56}$$

$$= 2[p(0) + c]\delta(\ln^2 \delta - 2\ln \delta + 2) < \infty. \tag{57}$$

The first and the third term can also be bounded. Since $\ln|x|$ is a convex function in the regions $x \geq \delta$ and $x \leq \delta$ respectively. We can find linear functions $ax + b$ of $x$ such that $ax + b \geq \ln x$ for $x \geq \delta$. This leads to

$$\int_{\delta}^{\infty} dx\ p(x)\ln^2|x| \leq \int_{\delta}^{\infty} dx\ p(x)(ax + b)^2 \tag{58}$$

$$\leq \int_{-\infty}^{\infty} dx\ p(x)(ax + b)^2 \tag{59}$$

$$= \int_{-\infty}^{\infty} dx p(x)(a^2x^2 + 2abx + b^2) \tag{60}$$

$$= a^2\mu_2 + 2ab\mu_1 + b^2, \tag{61}$$

where we have used the notation $\mathbb{E}[x^2] = \mu_2$ and $\mathbb{E}[x^1] = \mu$, which by assumption is finite. The case for $x \le -\delta$ is completely symmetric and can also be bounded by $a^2\mu_2 + 2ab\mu_1 + b^2$. Therefore, we have shown that

$$\mathbb{E}_{x \sim p(x)}[\ln|x|^2] \le 2(a^2\mu_2 + 2ab\mu_1 + b^2) + 2(p(0) + c)\delta(\ln^2\delta - 2\ln\delta + 2) \le \infty. \tag{62}$$

This proves the proposition. $\square$

**Remark.** *Note that, the continuity in the neighborhood of $0$ is not a necessary condition. For example, one can also prove that $\mathbb{E}_x[\ln^2|x|]$ is finite if $p(x)$ is bounded and its second moment exists.*

### C.2   PROOF OF PROPOSITION 2

*Proof.* The case when $w_0 = 0$ is trivially true, we therefore consider the case $w_0 \ne 0$. By Lemma 1, we have

$$\frac{1}{\sqrt{t}} z_t \to_d \mathcal{N}(0, s^2) \tag{63}$$

where we have defined $z_t = \frac{1}{\sqrt{t}}(\ln\left|\frac{w_t}{w_0}\right| - t\mu)$, $\mu = \mathbb{E}[\ln|1 - \lambda x|]$ and $s^2 = \mathrm{Var}[\ln|1 - \lambda x|]$. This means that

$$\lim_{t \to \infty} p\left(\frac{1}{\sqrt{t}}\ln\left|\frac{w_t}{w_0}\right|\right) = \lim_{t \to \infty} \frac{1}{\sqrt{2\pi s^2}}\exp\left[-\frac{1}{2ts^2}\left(\ln\left|\frac{w_t}{w_0}\right| - t\mu\right)\right]. \tag{64}$$

By definition, $|w_t| = |w_0|e^{\sqrt{t}z + t\mu}$. Therefore, we have

$$\lim_{t \to \infty} P(|w_t| > \epsilon) = \lim_{t \to \infty} P\left(|w_0|e^{\sqrt{t}z + t\mu} > \epsilon\right) = \begin{cases} 0 & \text{if } \mu < 0; \\ 1 & \text{if } \mu > 0, \end{cases} \tag{65}$$

for all $\epsilon > 0$. In other words, since $\mu$ and $\sigma$ are $t$-independent, in the infinite $t$ limit, the sign of $\mu$ becomes crucial. When $\mu > 0$, the limiting distribution diverges to infinity; when $\mu < 0$, $|w_t|$ converges to $0$ in probability. This finishes the proof. $\square$

### C.3   PROOF OF COROLLARY 1

*Proof.* By the definition of $p(x)$,

$$\mu = \mathbb{E}[\ln|1 - \lambda x|] \tag{66}$$

$$= \frac{1}{2}\ln|1 - \lambda| + \frac{1}{2}\ln|1 - \lambda(a - 1)| \tag{67}$$

$$= \frac{1}{2}\ln|(1 - \lambda)(1 - \lambda(a - 1))|, \tag{68}$$

while $s^2$ is indeed constant in time. Therefore, the asymptotic distribution of $w_t$ is solely dependent on the sign of $\mu$. The above equation implies that $\mu \ge 0$ when

$$|(1 - \lambda)[1 - \lambda(a - 1)]| > 1. \tag{69}$$

When $\lambda \le 1$, the above equation is solved by

$$\lambda < \frac{a}{a - 1}. \tag{70}$$

When $\lambda > 1$, the above equation solves to

$$\lambda \ge \frac{a - \sqrt{a^2 - 8a + 8}}{2(a - 1)}, \tag{71}$$

only when the learning rate satisfies the above two conditions can it escape from the local maximum. Conversely, SGD cannot escape the local minimum when

$$\frac{a}{a - 1} \le \lambda \le \frac{a - \sqrt{a^2 - 8a + 8}}{2(a - 1)}. \tag{72}$$

We are done. $\square$

## C.4 PROOF OF PROPOSITION 3

*Proof.* By Lemma 1, we have that

$$\frac{1}{t}\mathbb{E}\left[\ln\left|\frac{w_t}{w_0}\right|\right] = \mu = \frac{1}{2}\ln\{(1-\lambda)[1-\lambda(a-1)]\}, \tag{73}$$

where the second equality follows from the assumption that $\lambda < 1$. We differentiate with respect to $\lambda$ to find the critical escape rate:

$$\lambda^* = \frac{a}{2(a-1)}. \tag{74}$$

Since $\mu$ is convex in $\lambda$, it follows that this critical escape rate is the maximum escape rate. Plugging this into $\mu$, we obtain that

$$\gamma^* = \mu(\lambda^*) = \frac{1}{2}\ln\frac{(2-a)^2}{4(1-a)} = \ln\frac{2-a}{2\sqrt{1-a}} \le \epsilon, \tag{75}$$

i.e., the optimal escape rate can be made smaller than any $\epsilon$ if we set

$$|a| \le 2\left|-e^\epsilon\sqrt{e^{2\epsilon}-1} - e^{2\epsilon} + 1\right|. \tag{76}$$

This finishes the proof. □

## C.5 PROOF OF PROPOSITION 4

*Proof.* It suffices to show that $w_1$ converges to $0$ with probability $1$ because, if this is the case, the only possible local minimum to converge to is $(w_1, w_2) = (0, \pm\sqrt{\frac{1+a}{2}})$.

The SGD dynamics is

$$\begin{cases} \Delta w_{1,t} = -\lambda(-2w_{1,t} + 4w_{1,t}^3 + 12w_{2,t}^2 w_{1,t} + x_t b w_{1,t}); \\ \Delta w_{2,t} = -\lambda(-2w_{2,t} + 4w_{2,t}^3 + 12w_{1,t}^2 w_{2,t} - 2aw_{2,t}). \end{cases} \tag{77}$$

We focus on the dynamics of $w_1$. By the definition of the noise $x$, we have that

$$w_{1,t+1} = \begin{cases} w_{1,t}[1 + \lambda(2-b) - 4\lambda w_{1,t}^2 - 12\lambda w_{2,t}^2] & \text{with probability } 1/2; \\ w_{1,t}[1 + \lambda(2+b) - 4\lambda w_{1,t}^2 - 12\lambda w_{2,t}^2] & \text{with probability } 1/2. \end{cases} \tag{78}$$

Equivalently,

$$\left|\frac{w_{1,t+1}}{w_{1,t}}\right| = \begin{cases} \left|1 + \lambda(2-b) - 4\lambda w_{1,t}^2 - 12\lambda w_{2,t}^2\right| & \text{with probability } 1/2; \\ \left|1 + \lambda(2+b) - 4\lambda w_{1,t}^2 - 12\lambda w_{2,t}^2\right| & \text{with probability } 1/2. \end{cases} \tag{79}$$

Since $0 \le 4\lambda w_{1,t}^2 + 12\lambda w_{2,t}^2 \le 16\lambda$, we can define a new random variable $r_t$:

$$r_t := r(x_t) := \begin{cases} \max(|1 + \lambda(2-b)|, |1 + \lambda(2-b) - 16\lambda|) & \text{if } x_t \ge 0; \\ \max(|1 + \lambda(2+b)|, |1 + \lambda(2+b) - 16\lambda|) & \text{if } x_t \le 0. \end{cases} \tag{80}$$

By construction, $r_t \ge |w_{1,t+1}/w_{1,t}|$ for all values of $x_t$. This implies that $|w_{1,t}/w_{1,0}| \le \prod_{i=1}^t r_t$, and so,

$$P(|w_{1,t}/w_{1,0}| > \epsilon) \le P\left(\left|\prod_{i=1}^t r_t\right| > \epsilon\right), \tag{81}$$

i.e., if $\prod_{i=1}^t r_t$ converges to $0$ with probability $1$, $w_{1,t}$ must also converge to zero with probability $1$.

We let $b = \frac{1}{\lambda} - 6$ (note that this is the value of $b$ such that $r_t$ is minimized for both cases), and we obtain that

$$r_t = \begin{cases} 8\lambda & \text{if } x_t \ge 0; \\ 2\max(|1-2\lambda|, |1-10\lambda|) & \text{if } x_t \le 0. \end{cases} \tag{82}$$

The rest of the proof follows from applying the central limit theorem to $\frac{1}{\sqrt{t}}\sum_{i=1}^t \ln r_t$ as in Lemma 1. The result is that $\prod_{i=1}^t r_i$ converges to $0$ with probabiliy $1$ if

$$\mu := \mathbb{E}[\ln r_t] = \frac{1}{2}\ln[8\lambda\max(|1-2\lambda|, |1-10\lambda|)] < 0. \tag{83}$$

The above inequality solves to

$$\lambda \le \frac{1}{20}(1+\sqrt{6}) \tag{84}$$

which is of order $O(1)$ as stated in the theorem statement. This proves the proposition. □

### C.6 PROOF OF PROPOSITION 5

*Proof.* First we note that, since $|w_t| \le 1$

$$|\hat{g}_t| = |x_t w_t| \le 1 + a. \tag{85}$$

By the definition of the AMSGrad algorithm, we have that

$$v_t = \beta_2 v_{t-1} + (1 - \beta_2)\hat{g}_t^2; \tag{86}$$
$$\hat{v}_t = \max(\hat{v}_{t-1}, v_t). \tag{87}$$

Since $v_0 = 0$ and $|\hat{g}_t| = |x_t w_t| \le 1 + a$, we have that

$$v_t \le 1 + a \tag{88}$$

for all $t$, and so

$$\hat{v}_t \le \max_t \hat{v}_t = \max_t v_t \le 1 + a. \tag{89}$$

Meanwhile, since $\hat{v}_t$ is a monotonically increasing series and is upper bounded by $1 + a$, it must converge to a constant $0 < c \le 1 + a$.

Now, as before, we want to upper bound the random variable

$$\frac{1}{\sqrt{t}} \left( \ln \left| \frac{w_{t+1}}{w_t} \right| - \mu \right) = \frac{1}{\sqrt{t}} \sum_{i=1}^{t} \left( \ln \left| 1 - \frac{\lambda}{\sqrt{\hat{v}_t}} x_t \right| - \mu \right), \tag{90}$$

where $\mu = \mathbb{E}[\ln \left| \frac{w_{t+1}}{w_t} \right|]$. Since $\hat{v}_t$ converges to $c$, there must exists a positive integer $N(\epsilon)$ such that for any $\epsilon > 0$, $\hat{v}_N \le c - \epsilon$. This allows us to divide the sum to two terms:

$$\frac{1}{\sqrt{t}} \left( \ln \left| \frac{w_{t+1}}{w_t} \right| - \mu \right) = \frac{1}{\sqrt{t}} \sum_{i=1}^{t} \left( \ln \left| 1 - \frac{\lambda}{\sqrt{\hat{v}_t}} x_t \right| - \mu \right) \tag{91}$$

$$= \frac{1}{\sqrt{t}} \sum_{i=1}^{N} \left( \ln \left| 1 - \frac{\lambda}{\sqrt{\hat{v}_t}} x_t \right| - \mu \right) + \frac{1}{\sqrt{t}} \sum_{i=N+1}^{t} \left( \ln \left| 1 - \frac{\lambda}{\sqrt{c - k_t}} x_t \right| - \mu \right), \tag{92}$$

where we introduced $0 \le k_t \le \epsilon$. But the first term is of order $O(1/\sqrt{t})$ and converges to 0, and so for sufficiently large $t$, the first term is also smaller than arbitrary $\epsilon$. This means that

$$\frac{1}{\sqrt{t}} \left( \ln \left| \frac{w_{t+1}}{w_t} \right| - \mu \right) \le \frac{1}{\sqrt{t}} \sum_{i=1}^{t} \left( \ln \left| 1 - \frac{\lambda}{\sqrt{c - k_t}} x_t \right| - \mu \right) + \epsilon, \tag{93}$$

We can now consider the random variable

$$\left| 1 - \frac{\lambda}{\sqrt{c - k_t}} x_t \right| = \begin{cases} \left| 1 - \frac{\lambda}{\sqrt{\hat{v}_t}} \right| & \text{with probability } \frac{1}{2}; \\ 1 + \frac{\lambda}{\sqrt{\hat{v}_t}}(1 - a) & \text{with probability } \frac{1}{2}. \end{cases} \tag{94}$$

We now define a new random variable

$$r_t(x_t) := \begin{cases} m & \text{if } x_t = 1; \\ 1 + \frac{\lambda}{\sqrt{c - \epsilon}}(1 - a) & \text{if } x_t = -1 + a. \end{cases} \tag{95}$$

where we defined the constant $m := \max(|1 - \lambda/\sqrt{c}|, |1 - \lambda/\sqrt{c - \epsilon}|)$. One can easily check that $r_t \ge \left| 1 - \frac{\lambda}{\sqrt{c - k_t}} x_t \right|$. This means that

$$P\left( \left| \frac{w_{t+1}}{w_0} \right| > \alpha \right) \le P\left( \prod_{i=0}^{t} r_i > \alpha + O(1/\sqrt{t}) \right), \tag{96}$$

i.e., if $r_t$ converges to 0 in probability, $\left| \frac{w_{t+1}}{w_0} \right|$ must also converge to 0 in probability. Proceeding as in the proof of Proposition 2. One finds that the condition for $r_t$ to converge in probability to 0 is

$$\ln \left| m \left[ 1 + \frac{\lambda}{\sqrt{c - \epsilon}}(1 - a) \right] \right| < 0, \tag{97}$$

which is equivalent to

$$\left| m \left[ 1 + \frac{\lambda}{\sqrt{c}} (1 - a) \right] \right| < 1. \tag{98}$$

Since $\epsilon$ is arbitrary, we let $\epsilon \to 0$ and obtain

$$\left| \left( 1 - \frac{\lambda}{\sqrt{c}} \right) \left[ 1 + \frac{\lambda}{\sqrt{c}} (1 - a) \right] \right| \leq 1. \tag{99}$$

Denote $\lambda/\sqrt{c}$ as $\lambda'$, this condition solves to

$$\frac{a}{a - 1} \leq \lambda' \leq \frac{a - \sqrt{a^2 - 8a + 8}}{2(a - 1)}. \tag{100}$$

Setting $a$ such that the above condition is met, AMSGrad will converge to $0$ in probability. For $\lambda < 1$,

This completes the proof. □