# OpenReview forum: "SGD Can Converge to Local Maxima"
_ICLR.cc/2022/Conference — ICLR 2022 Spotlight_

### Official Review · Reviewer_bMKE · 2021-10-21

**Correctness:** 3
**Technical Novelty And Significance:** 4
**Empirical Novelty And Significance:** 2
**Recommendation:** 8
**Confidence:** 3

**Main Review:**

This paper brings a very interesting viewpoint on the optimization process of SGD. I think it is widely conjectured that adding noise to the optimization algorithm is beneficial (e.g. Jin et al. 2017). In this paper, several artificial examples are given where adding noise can have the opposite effect, e.g. SGD may converge to a global maximum while using GD on the population loss of the same objective will not. This raises many interesting questions regarding the effect of noise in optimization and is a novel contribution as I understand it.

The authors also provide several different examples for which SGD behaves in an unintuitive way, e.g. converging to a sharp minimum rather than a flat one. This may show that the results for implicit bias (which are usually given for GD or GF) may not be directly applicable to SGD. The experimental section is also nice and provides empirical evidence for the validity of the theoretical results. Specifically, the one neuron example provides evidence that when the assumptions from previous papers on escaping saddle points are not satisfied, then it may happen that indeed SGD may not escape them.

There is one important issue which I would be happy to see the author’s response:

The difference between the results in Section 5.2 and 5.4 is confusing. In 5.2 the claim is that even if the learning rate changes, then SGD may escape saddle points slowly. In Proposition 3 there doesn’t seem to be any sign that the learning rate can change. If it doesn’t change during training, then how is this result different from the one given in 5.1?
If the learning rate does change, then in Section 5.4, AMSGrad without momentum is just SGD with an adaptive learning rate, how is it different from the result in 5.2?

There are also some issues with the presentation, but  I believe they can be fixed:

1) What is \delta in Proposition 3 and Corollary 1? The only definition of \delta I found was in Lemma 1 in the appendix. If it is really as defined there then it is extremely confusing and should be defined in the main part of the paper.

2) In Definition 3, why is w_t inside ln? Could it just be defined without ln? I think the authors should provide some intuition for this definition.

3) Having both Theorem 1 and Proposition 3 is redundant. I think it would be clearer to just write the main result here.

4) Also, Theorem 1 talks about *the* saddle point, while from the phrasing of the theorem it is not clear why there is a single saddle point.

5) Figure 2 is not a figure, but an algorithm (or just a list of equations).

6) Beginning of Section 6: what the underline should highlight? It breaks in a middle of a word…

7) The d subscript on the arrow (e.g. equations 59, 62). Is it convergence with probability? I think this is a non-standard notation and should be defined.


**Summary Of The Paper:**

The paper provides several artificial examples on which SGD has an unintuitive behavior. This includes: (1) SGD converges to a local maximum; (2) If the learning rate is not fixed then SGD takes arbitrarily long time to escape saddle points; (3) SGD may prefer sharper minima (in contrast to several hypotheses regarding the implicit bias of GD); (4) Adaptive methods may also converge to a global maximum. Several experiments are made, varifying the theoretical results.

**Summary Of The Review:**

I think this paper provides an interesting and novel view on SGD, and brings forward scenarios for which SGD behaves differently from GD and from the intuition provided in previous papers. This is an important step in understanding the optimization process of SGD, its limitations, and its benefits. There is one issue which I would be happy to see the author’s response. There are also some issues with the presentation which I think can be fixed. In total, I believe the merits of this paper outweigh its flow and it should be accepted.

---

> ### Author Response · Authors · 2021-11-19
> **Reply**
>
> *Q1: “There is one important issue which I would be happy to see the author’s response: The difference between the results in Section 5.2 and 5.4 is confusing. In 5.2 the claim is that even if the learning rate changes, then SGD may escape saddle points slowly. In Proposition 3 there doesn’t seem to be any sign that the learning rate can change. If it doesn’t change during training, then how is this result different from the one given in 5.1? If the learning rate does change, then in Section 5.4, AMSGrad without momentum is just SGD with an adaptive learning rate, how is it different from the result in 5.2?”*
>
> * Thanks for your question, and we apologize for causing the confusion. This confusion is due to a misunderstanding of the word “change.” By the word “changing,” we do not mean that the learning rate is changing during training, but that the (hypothetical) experimenter is able to observe the local landscape and the noise and is capable of setting the learning rate to be any desired (but fixed) value before training. Once set, the learning rate is not allowed to change during the training. By doing this, the experimenter gains the advantage of being able to use an optimal “a posteriori” learning rate, and Section 5.4 shows that even if the experimenter can find the optimal fixed learning rate, the escape can be arbitrarily slow.
> * In contrast, Section 5.2 is more “adversarial” towards the hypothetical experimenter. In this experiment, the landscape is adversarially chosen against the fixed learning rate of the experimenter. In summary, from the perspective of the experimenter, section 5.2 studies his or her worst-case performance, and section 5.4 studies the best-case performance.
> * As described above, since the learning rate is not changing as your question assumes, the question about AMSGrad does not seem to be relevant.
>
> *Q2: What is \delta in Proposition 3 and Corollary 1? The only definition of \delta I found was in Lemma 1 in the appendix. If it is really as defined there then it is extremely confusing and should be defined in the main part of the paper.*
> * Sorry for not clearly defining $\delta$. In the main text, $\delta(x)$ is the Dirac delta function. We will add this statement to the main text.
>
> *Q3: In Definition 3, why is w_t inside ln? Could it just be defined without ln? I think the authors should provide some intuition for this definition.*
> * Thanks for the suggestion. This definition is motivated by the proof of proposition 2. For the example we considered, whether the escape is successful crucially depends on the sign of $\mathbf{E}[\ln w_t]$; see the proof sketch of proposition 2 for a discussion.
> * Another intuition comes directly from the escaping problem of dynamical systems in continuous time. In continuous-time, the equation of motion for the parameter $w$ is often solved to be $e^{c_0t}$, where the exponent coefficient $c_0$ is called the escape rate: the larger the $c_0$, the faster it takes to escape to infinity. Definition 3, therefore, approximately extracts the $c_0$ factor from the trajectory of $w$.
>
> *Q4: Having both Theorem 1 and Proposition 3 is redundant. I think it would be clearer to just write the main result here.*
> * Thanks for the suggestion. We updated Section 5.2 to combine the two theorems to reduce redundancy.
>
> *Q5: Also, Theorem 1 talks about the saddle point, while from the phrasing of the theorem it is not clear why there is a single saddle point.*
> * The updated statement is now clearer.
>
> *Q6: Beginning of Section 6: what the underline should highlight? It breaks in a middle of a word…*
> * Sorry, this is not an underline but the paper format for separating the footnote and the main text. Since this problem follows from the ICLR stylefile, it is hard for us to fix it directly.
>
> *Q7: The d subscript on the arrow (e.g. equations 59, 62). Is it convergence with probability? I think this is a non-standard notation and should be defined.*
> * This notation means convergence in distribution. We added the definition in the manuscript.

---

> > ### Comment · Reviewer_bMKE · 2021-11-28
> > **Re: Reply**
> >
> > Thank you for the clarifications and the updated manuscript, they addressed my concerns and comments. I will maintain my score.

---

### Official Review · Reviewer_MX9i · 2021-10-24

**Correctness:** 4
**Technical Novelty And Significance:** 2
**Empirical Novelty And Significance:** 2
**Recommendation:** 6
**Confidence:** 4

**Main Review:**

This paper is well written and has a good presentation. The work constructs some examples to illustrate the possible counter-intuitive phenomena of SGD. While the main point of the paper is very interesting, it is not clear what can we learn from these special example problems.

Specifically, the example in eq.(2) constructed to show convergence to local maxima has a very special structure that is unlikely to occur in machine learning practice. The main point conveyed by the authors is that the population loss associated with eq(2) is a strongly concave function with a local maxima point at $w=0$, and if we perform SGD with the constructed Bernoulli data distribution, then it is possible that SGD converges to this local maxima w.p. 1. However, the constructed Bernoulli data distribution generates negatively correlated data samples, i.e., $x = 1$ or $-1+a, a<0$. This is the main reason that SGD may get stuck at this local maxima, because the loss associated with the data $x=1$ has a strongly convex landscape while the data $x=-1+a$ has a strongly concave landscape. Hence, one can find a regime of learning rate in which the gradient descent on the strongly convex loss dominates the strongly concave loss, and converges to the maxima. However, this requires the data points are highly negatively correlated, which rarely happens in machine learning practice.

The other constructed examples in the paper have similar issues. Overall, the examples given in the paper may be too extreme to provide a good understanding of SGD in machine learning practice. On the other hand, this may suggest that in addition to the existing assumptions of SGD, there might be other undiscovered structures in deep learning that prevent SGD from behaving anomalously.


**Summary Of The Paper:**

Many theoretical works have studied SGD, but they commonly rely on
restrictive and unrealistic assumptions about the noise. In this work,
the authors construct example optimization problems illustrating that, if these assumptions
are relaxed, SGD can exhibit many strange behaviors, including (1) SGD can converge to
local maxima, (2) SGD may escape saddle points arbitrarily slowly, (3) SGD
can prefer sharp minima over flat ones, and (4) AMSGrad can converge to local
maxima.
Therefore, the authors conclude that in the most general nonconvex case, many counter-intuitive phenomena of SGD may arise and
contrast to the commonly held presumptions.

**Summary Of The Review:**

See the comments above.

---

> ### Author Response · Authors · 2021-11-19
> **Reply**
>
> Thank you for your review. We agree with your criticism that the example we provided is special; this has also been pointed by other reviewers. However, we would like to convince you that the fact that our construction is special does not reduce its theoretical value for the community.
>
> *Q1: “While the main point of the paper is very interesting, it is not clear what can we learn from these special example problems.”*
> * We believe that there are many insights and implications of our results, including one implication identified by you. Let us summarize the novel lessons of this work that have either been stated in the manuscript or pointed out by other reviewers, and we hope that we can convince you to reconsider the significance of the present work:
> 1. “The paper has explicitly demonstrated that the convergence point in expectation is distinct from that with high probability. In practice, people care more about convergence with high probability. Thus, this paper emphasizes the importance of convergence with high probability.” (reviewer k2vd and DCLz)
> 2.  “In addition to the existing assumptions of SGD, there might be other undiscovered structures in deep learning that prevent SGD from behaving anomalously.” (You)
> 3. Our work offers an important cautionary note regarding the study of saddles point in deep learning: “The fundamental concept it suggests seems to be that we need to think more carefully about our assumptions and goals when proving results about optimization algorithms - e.g. it is often believed that convergence to critical points is fine since very likely adding a little noise will prevent converging to maxima or escaping saddle points.” (reviewer Jqcm)
> 4. The importance of the noise for understanding SGD: “This may show that the results for implicit bias (which are usually given for GD or GF) may not be directly applicable to SGD.” (reviewer bMKE)
> * Along with the four messages stated in the abstract and their respective implications:
> 5. SGD can converge to a local maximum
> 6. SGD might only escape saddle points arbitrarily slowly -- this implies that escaping rate from a saddle point is not a monotonic function of the learning rate, which has been conventionally thought to be the case
> 7. SGD can prefer sharp minima over flat ones -- which shows that a series of recent theoretical works that argue SGD prefer flatter minima may need some level of reconsideration (see the related works given in Section 5.3)
> 8. AMSGrad converge to a saddle point -- this is also very important for the field of deep learning because of the prominence of the adaptive gradient methods in practice and the widely accepted importance of their convergence properties
> * In short, we believe that there are many important aspects of deep learning optimization that this works can shed light on, and we hope that you might reconsider your criticism of “it is not clear what can we learn from these special example problems.”
>
>
> *Q2: “...Hence, one can find a regime of learning rate in which the gradient descent on the strongly convex loss dominates the strongly concave loss, and converges to the maxima. However, this requires the data points are highly negatively correlated, which rarely happens in machine learning practice.”*
>
> * This statement is incorrect. The data points in real problems can easily be highly negatively correlated. As the neural example in section 6.3 shows, the gradient from data points can easily be negatively correlated when the data points contain inherent uncertainty or label noise, which is well known to happen for various realistic settings. For example, see this work: https://arxiv.org/pdf/1904.06963.pdf.

---

### Official Review · Reviewer_k2vd · 2021-11-01

**Correctness:** 3
**Technical Novelty And Significance:** 3
**Empirical Novelty And Significance:** 2
**Recommendation:** 8
**Confidence:** 4

**Main Review:**


The abnormal behavior of SGD is out of expectation from the first sight. At the same time these behaviors should be as expected because the experimental settings are specifically chosen and break usual assumptions.

Strong points.
The paper has explicitly demonstrate the convergence point in expectation is distinct from that with high probability. In practice, people care more about the convergence with high probability. Thus, this paper emphasizes the importance of the convergence with high probability.
The paper questions the usual assumptions and gives convincing theoretical and empirical evidence. This view of research is encouraging.
The paper is clear and easy to follow. The reviewer verified several claims and did not find wrong points, though the whole proof was not thoroughly examined.

Weak points.
The artificial examples are far from the optimization of practical neural network. To make the result more convincing and useful in practice, it still needs to study the with high probability of the model setting and data setting, how SGD will perform and whether the constructed scenarios will be encountered.

**Summary Of The Paper:**


The paper constructs example optimization problems to illustrate that SGD may behave strangely, out of common expectation, when some assumption is removed, e.g., decaying learning rate and the noise nature. It shows that SGD can converge to local maxima with high probability for specific cases.

**Summary Of The Review:**

The paper has interesting observations including  for SGD, which are not carefully discussed in literature. Personally the reviewer would like to see the paper published on important venue like ICLR.

---

> ### Author Response · Authors · 2021-11-19
> **Reply**
>
> Thank you for your review.
>
> *Q1: “Weak points. The artificial examples are far from the optimization of practical neural network. To make the result more convincing and useful in practice, it still needs to study the with high probability of the model setting and data setting, how SGD will perform and whether the constructed scenarios will be encountered.”*
> * Thanks for raising this. We agree that this is a limitation of the present work and, as commented in our conclusion, we believe it is an important future direction to prove or disprove this phenomenon in more realistic settings.

---

### Official Review · Reviewer_Jqcm · 2021-11-02

**Correctness:** 4
**Technical Novelty And Significance:** 4
**Empirical Novelty And Significance:** 4
**Recommendation:** 8
**Confidence:** 4

**Main Review:**

I found this paper thought-provoking and enjoyable to read. The fundamental concept it suggests seems to be that we need to think more carefully about our assumptions and goals when proving results about optimization algorithms - e.g. it is often believed that convergence to critical points is fine since very likely adding a little noise will prevent converging to maxima or escaping saddle points.

This leads to my first objection: the theoretical results would be much improved if they included an analysis of adding a small amount of noise to SGD's iterates. However, from looking at the constructions my intuition is that such perturbations may not change the overall message of the paper too much.

Another missing component is an analysis of the "theoretical standard" decaying learning rate of $O(1/\sqrt{t})$ (note that AMSgrad does not have this decaying property).

My main concern thus is that the paper is perhaps not going the full distance to realize its goals, and I would encourage the authors to address these issues in the revision. However, I think overall the observations are useful and may inform future thought about design of optimization algorithms or analysis.

Note: for proposition 3, I believe do you need to say that $a<0$?

**Summary Of The Paper:**

This paper demonstrates on several fairly simple (e.g. 1-dimensional quadratic) objectives that stochastic gradient descent may easily have very poor behavior: it could converge to a maximum, or diverge even in convex settings if the learning rate is too high.
Specifically, it is shown that for any learning rate, there is is a distribution over quadratics whose expectation is $-rx^2$ for some $r\ge 0$ such that SGD will converge to the maximum at 0, and there is also a distribution whose expectation is $rx^2$ such that SGD still diverges even on this convex loss. Note that the distribution (and $r$) depend on the learning rate.
Further, there are distributions such that SGD must escape a saddle point very slowly, and a distribution over 2-dimensional quartics such that SGD will converge to a "sharp" rather than a "flat" minimum.
It is also shown that AMSgrad must converge to a local maximum on some non-convex distributions.

The results are proven by choosing a particular distribution over quadratic objectives that causes the logarithm SGD's iterate to be a sum of i.i.d. random variables. Then the central limit theorem provides an understanding of the limiting behavior of these iterates.

The results are augmented by empirical studies verifying the theorems directly, along with a very simple small neural network experiment.

**Summary Of The Review:**

The paper introduces some interesting examples accompanied with relevant commentary. The examples are simple enough to be easily understood, but may leave a bit to be desired in generality.

---

> ### Author Response · Authors · 2021-11-19
> **Reply**
>
> *Q1: “…This leads to my first objection: the theoretical results would be much improved if they included an analysis of adding a small amount of noise to SGD's iterates. However, from looking at the constructions my intuition is that such perturbations may not change the overall message of the paper too much.”*
> * Thanks for mentioning this very important comparison (escaping with vs. without injected noise). In our manuscript, we did perform a continuous-time analysis of this case in the appendix. See the experiment in Figure 10 and the associated analysis in section C.2. The numerical simulation and the analysis show that for the example we considered, even adding in additive noise still does not help escaping -- the stationary distribution is no longer a delta distribution but still confined around the saddle point. This comparison strengthens the conclusion made in the main text -- even adding artificial perturbations cannot help with escaping.
>
> *Q2: “Another missing component is an analysis of the "theoretical standard" decaying learning rate of  $O(1/\sqrt{t})$ (note that AMSgrad does not have this decaying property).”*
> * Thanks for mentioning. Indeed, we do not consider the situation that the learning rate decays as $O(1/\sqrt{t})$. We think this is valuable future work to do but beyond the scope of the present work.
>
> *Q3: “for proposition 3, I believe do you need to say that $a<0$?”*
> * Correct. We will add this.

---

### Official Review · Reviewer_DCLz · 2021-11-03

**Correctness:** 3
**Technical Novelty And Significance:** 4
**Empirical Novelty And Significance:** 2
**Recommendation:** 6
**Confidence:** 5

**Main Review:**

## Post-discussion reassessment

See relevant post below.

## Evaluation and recommendation

There are two axes that have to be evaluated in this paper: the mathematical value of the authors' contributions, and the positioning thereof. Regrettably, my assessment varies drastically along these two axes: while I find the authors' results mathematically interesting and suitable for ICLR, the overall narrative is encumbered by a series of vague and often confusing statements that are (a) detrimental to the paper; and (b) not needed in the first place. Thus, while on the mathematical axis I consider the paper to be a good fit for ICLR (I would rate it around a 7), the reported results do not suffice to support the paper's (overly ambitious) narrative claims, hence my overall reject recommendation.

I believe that these shortcomings can be mitigated by toning down the paper's claims - but this will also require significant work from the authors. For this reason, I am providing below some necessary context for item (1) above (which, judging from the paper's title, seems to be what the authors consider to be the main contribution of the paper).

To begin, if we consider ordinary (non-stochastic) gradient descent applied to the loss function $L(w) = w^2/2$, it is trivial to see that the generated sequence explodes in value if $\lambda>1$ – perhaps the worst possible "convergence to a global maximum" catastrophe that one could encounter. This is a fundamental and inescapable failure of gradient descent: if the step-size is not chosen appropriately, all sorts of undesirable phenomena can be observed (both in practice and in theory). Well-documented failures of this type are precisely the reason that the tuning of the step-size of SGD plays such an important role in the field. Hence, in this context, the authors' statement that undesirable phenomena (convergence to maximizers) can indeed occur if SGD is run with a large, out-of-tune step-size is hardly "surprising". [To be clear, I _do not_ believe that a result needs to be "surprising" in order to be worthwhile, but I _did_ lose count of how many times the authors claimed their results are "surprising".]

Now, what is interesting in the authors' analysis is the specification of the range of $\lambda$ for which these deleterious outcomes can arise in the quadratic toy model of item (1) above and the CLT analysis that they provide (which, incidentally, relies heavily on the specific loss model considered). However, what lessons can be inferred for neural network training from this 1-dimensional example is highly debatable - and the authors' one-neuron, quadratic-activated "neural-network-like" example in Section 6.3 is no less artificial than the examples of Sections 4 and 5. [By the way, given that both of the authors' examples are custom-made to allow calculations in quasi-closed form, statements concerning "restrictive and unrealistic assumptions about the nature of the noise" seem out of place.]

What I found lacking in the above is a thorough comparison with the work of Ge et al. [1]. Specifically, how does the authors' "trapped regime" compare to that predicted by [1]? More concretely, if we take Corollary 1 as a starting point, what is the precise value for $\lambda$ below which [1] guarantees saddle-point avoidance? Is there a gap with the authors' trapping bounds or are they sharp? Given that the two results – that of the authors' and that of [1] – indicate completely different behaviors for (SGD), I would expect a much more in-depth comparison between the two.

In a similar vein, it is also unfortunate that the authors seem to ignore a series of results showing that SGD with a decreasing step-size avoids saddle-points with probability $1$ – cf. the general paper [2] and the more recent references [3,4] below. In fact, one of the most popular neural network training methods involves staircase step-size schedules that begin at a large value which is then progressively halved towards the end of the training horizon - and this, precisely to enhance the algorithm's convergence properties. The authors' theory does not cover this important part of the literature, so their claims are - at best - incomplete in this regard.

To summarize, the paper does not "run counter to the established wisdom of the field". If anything, it serves to _reinforce_ this wisdom by providing an interesting cautionary tale to the effect that "SGD _with a large, constant step-size_ can converge to local maxima".

If the authors can provide a revised version toning down the various overselling issues identified above, I would be happy to raise my score accordingly. Otherwise, even though I find the paper's mathematical contributions interesting and suitable for ICLR, I cannot recommend acceptance at this stage: the paper contains a number of "sound bite"-like statements that are non-mathematical and not appropriate for a theory paper, so this would only serve to increase confusion in the field.


## Specific comments

To help the authors, I am providing below a list of specific points that should be revised to better reflect the paper's actual contributions.

1. Title: see above. Without any further quantifiers (and despite the word "can"), the current title suggests that "convergence to maximizers" is a ubiquitous phenomenon - whereas it is anything but.

1. Abstract: the statement concerning restrictive and unrealistic assumptions is out of place. If the authors refer to the "multiplicative" nature of the noise, they should note that the stochastic loss function $\hat L(w;x) = (w-x)^2/2$, $x\in\{-1,1\}$, has additive noise at its minimum and it is neither more nor less "realistic" than their toy example. In fact, it can be argued that the principal reason that the author's example exhibits convergence to a non-minimizing stationary point is that all batches become critical at the _same_ point - but, in turn, this is an unrealistic assumption in itself.

    At any rate, blanket statements like this should be avoided unless the authors are prepared to back them up with a deeper theoretical treatment.

1. A minor - and, admittedly, subjective - remark: the choice $a<0$ can become confusing in reading the various expressions (keeping track of what is negative and what isn't). I believe it would improve readability if $x$ took the values $1$ and $-1-a$ with $a>0$ (i.e., switching notation from $a$ to $-a$), but this is of course up to the authors.

1. A more important issue: the $a>0$ case is poorly explained in the paper and the corresponding part of Fig. 1 is misleading. For $a>0$, convergence to $0$ is the desired behavior, so coloring that region as "problematic" is not appropriate.

1. I was bemused with the "sharp vs. shallow" statement: especially since the trace of the Hessian is only a local attribute that does not suffice to characterize the basin of attraction of a (non-quadratic) minimum, I do not see why this is "counter to established wisdom to the field". Again, I would recommend providing an accurate mathematical description and letting readers draw their individual conclusions instead of trying to maneuver the narrative in this way.

1. The authors are making a series of claims for the applicability of their results to neural networks, but these are based on a "network" with a single neuron. I would again recommend moderation - otherwise, this is a textbook case of a faulty generalization.

1. The statement that "SGD noise is multiplicative and state-dependent" is too vague and lacks context. Again, I understand that the authors wish to motivate their specific counterexample, but blanket statements like that only serve to weaken their paper - not strengthen it.

1. When mentioning continuous-time approaches, I was surprised that the authors did not discuss the stohastic approximation framework for the study of SGD by Ljung, Kushner and Yin, and Pemantle (to state but some of the most classical results). I already mentioned this literature in the context of SGD with a decreasing step-size above, and since the authors seem to be interested in the continuous-time limit, this is a second reason why this literature should be discussed in detail. This also comes up when discussing related works in p.2 (as these works do not assume "artificially injected noise").

1. Still on the issue of related work: the authors seem to be confusing the Polyak-Łojasiewicz condition with the Kurdyka-Łojasiewicz condition. The former is global, and indeed restrictive; the latter is local, and includes all semi-algebraic functions (or more general any function defined by an o-minimal structure). I would encourage the authors to study in more depth the work of Bolte and co-authors on the topic - and, in addition, I would like to point out that all examples considered by the authors satisfy the KL condition.

1. Appendices B and C are quite disconnected from the rest of the paper: they concern a completely different model with very different results and attributes, so I would suggest removing them altogether.

1. While interesting and easy to work out, Proposition 1 is very special: if $\lambda=1$, we get an exact Newton update, which gets to $0$ in a single iteration, and thus remains there forever due to the authors' multiplicative randomness model. In this regard, Proposition 1 is not truly representative of what's going on (e.g., in Proposition 2).

1. In their discussion of Proposition 1, the authors also briefly discuss the difference between convergence in probability and convergence in expectation. I believe this distinction in the modes of convergence of random variables is an important take-away of this work, and one worth describing in more detail. If the paper's message is that we need to be careful about how we interpret SGD convergence and avoidance results, then this should be made clearer.

Overall, I trust that the above should suffice to indicate the tone and tenor of the changes that would be expected from a revision. As I said, the authors do not need to oversell their contributions: in my opinion, the mathematical content of the paper is enough for ICLR, and taking a more factual approach in describing said contributions would be much better for the paper.



## References

[1] Rong Ge, Furong Huang, Chi Jin, and Yang Yuan, Escaping from saddle points – Online stochastic gradient for tensor decomposition, COLT ’15: Proceedings of the 28th Annual Conference on Learning Theory, 2015.

[2] Robin Pemantle, Nonconvergence to unstable points in urn models and stochastic aproximations, Annals of Probability 18 (1990), no. 2, 698–712.

[3] Panayotis Mertikopoulos, Nadav Hallak, Ali Kavis, and Volkan Cevher, On the almost sure convergence of stochastic gradient descent in non-convex problems, NeurIPS ’20: Proceedings of the 34th International Conference on Neural Information Processing Systems, 2020.

[4] Stefan Vlaski and Ali H. Sayed, Second-order guarantees of stochastic gradient descent in non-convex optimization, https://arxiv.org/abs/1908.07023, 2019.

**Summary Of The Paper:**

## Post-discussion reassessment

See relevant post below.

## Summary of contributions

In this paper, the authors examine whether SGD with a large, constant step-size avoids local maximizers (or, more generally, undesirable saddle points of the underlying minimization problem). More precisely, they focus on the algorithm
$$
w_{t+1} = w_t - \lambda \hat g_t
$$
where $\lambda>0$ is the algorithm's step-size, and $\hat g_t$ is a (stochastic) gradient of the (stochastic) loss function $\hat L(w;x)$, with $x$ a random variable.

The paper's results can be summarized as follows:

1. If $\hat L(w;x) = (x/2) \cdot w^2$, the authors identify a range of values of $\lambda$ (which depends on the distribution of $x$) such that, in probability, $w_t$ converges to $0$ – which, under the specified distributional assumptions for $x$, is the global maximum of $L = \mathbb{E}[\hat L]$. This is made precise in Propositions 1 and 2, and Corollary 1.

2. They provide a quartic loss function under which SGD converges to the function's sharper minimizers (as measured by the trace of the Hessian at said points). [Proposition 4]

3. They provide a specific range of parameters under which the AMSGrad algorithm converges to the undesirable maximizer of item (1) above.

[In the supplement, the authors also provide an analysis of a gradient-like diffusion (Appendices B and C), which they discuss as a continuous-time model of (SGD). This part is not directly connected to the rest of the paper, so I am not including it in my evaluation below.]

**Summary Of The Review:**

There are two axes that have to be evaluated in this paper: the mathematical value of the authors' contributions, and the positioning thereof. Regrettably, my assessment varies drastically along these two axes: while I find the authors' results mathematically interesting and suitable for ICLR, the overall narrative is encumbered by a series of vague and often confusing statements that are (a) detrimental to the paper; and (b) not needed in the first place. Thus, while on the mathematical axis I consider the paper to be a good fit for ICLR (I would rate it between 6 and 7), the reported results do not suffice to support the paper's (overly ambitious) narrative claims, hence my overall reject recommendation.

The detailed scores below do not refer solely to the paper's technical proofs, but they also take into account the various non-mathematical claims made by the authors throughout the paper.

---

> ### Author Response · Authors · 2021-11-19
> **Reply part 1**
>
> Thank you for your detailed review. We carefully studied your review and agreed with part of your criticism. We reread our paper with your comments in mind, and we agree with your criticism that we should tone down some of our statements. We also adapted our manuscript in response to your review, mainly including three points: (1) toning down the claims/discussions in the narration, (2) discussion of the related works, and (2) a more detailed discussion of the role of multiplicative noise.
>
> However, we also disagree with part of your arguments after careful thought. We find that quite a few of your criticisms are likely due to your misunderstanding about the role of multiplicative noise in our theory -- we think it is our fault for having not sufficiently discussed the multiplicative noise in the main text that led to your misunderstanding. Also, the indispensable role of continuous-time analysis is also neglected in your review. In fact, we believe that it would be hard to fully understand our result without understanding the role of the multiplicative noise and continuous-time analysis we presented in the paper.
>
> *Note on the relevance of the continuous-time analysis and multiplicative noise*. To be specific, in all the four examples, it is the case that the noise in the gradient is multiplicative (namely, dependent on $w$) -- this multiplicative dependence is a direct consequence of the minibatch sampling technique (see the discussion at the end of Section C.4, for example).  If the multiplicative noise does not exist in these examples, SGD will never converge to the saddle point -- this point is crucially supported by the continuous-time analysis we offered in Section B and C. Claiming that section B and C are irrelevant and “disconnected” from the rest of the paper is incorrect.
>
> The continuous-time analysis has two more important purposes: (1) the upper branch of the escaping behavior (very large $\lambda$ in corollary 1) is not present in the continuous-time analysis, and we used this fact to argue that escaping with a very large learning rate is not desirable, which is in turn used to justify the conditions we assume for the later examples (see the discussion at the end of Section 5.1); (2) the continuous-time analysis further shows that, even if additive noise is added (as is suggested in previous works as a practical trick to escape saddle points), the stationary distribution is still confined close to the saddle point -- this point can only be theoretically understood in a continuous-time analysis. This result further suggests that the standard wisdom of injecting some noise helps escaping saddle points may not apply in general settings -- a crucial point that is also pointed out by reviewer Jqcm.
>
> Therefore, we have the feeling that you might have been slightly overconfident and somewhat subjective in your judgment. It does not seem reasonable that, without understanding two crucial analyses of the paper, the reviewer can claim to be “absolutely certain” about the review. In the following response to your questions, let us explain why we agree/disagree with the specific questions you raised. Your opinion is important to us, and we are curious about whether you agree with the points that we disagree with. We are still happy to make further revisions for the points we disagree with if you can convince us why our replies are not reasonable.

---

> > ### Author Response · Authors · 2021-11-19
> > **Reply part 2**
> >
> > *Q1: "To begin, if we consider ordinary (non-stochastic) gradient descent applied to the loss function $L(w) = w^2/2$, it is trivial to see that the generated sequence explodes in value if $\lambda > 1$ – perhaps the worst possible "convergence to a global maximum" catastrophe that one could encounter. This is a fundamental and inescapable failure of gradient descent: if the step-size is not chosen appropriately, all sorts of undesirable phenomena can be observed (both in practice and in theory). Well-documented failures of this type are precisely the reason that the tuning of the step-size of SGD plays such an important role in the field. Hence, in this context, the authors' statement that undesirable phenomena (convergence to maximizers) can indeed occur if SGD is run with a large, out-of-tune step-size is hardly "surprising". [To be clear, I do not believe that a result needs to be "surprising" in order to be worthwhile, but I did lose count of how many times the authors claimed their results are "surprising".]"*
> > * In retrospect, we agree that we used the word “surprising” too many times (9 in total); we thus carefully rewrote the sentences that contain the word “surprising”, and only use “surprising” for the most surprising results of ours (now only 3 times in total). However, we disagree with the argument that our result is “hardly surprising;” let us explain why.
> > * First of all, this criticism also misinterprets our result. Our result does not show that many surprising things arise only when the learning rate is large. In fact, what we have studied does not appear when one only has a large learning rate -- they only appear due to the crucial and curious interplay between a large learning rate and minibatch noise (see the comment above “Note on the relevance of the continuous-time analysis and multiplicative noise”). Therefore, this criticism does not seem to apply to our work.
> > * Moreover, the basis of this criticism is claimed without any reference/evidence. In particular, this criticism is based on the claim that “if the step-size is not chosen appropriately, all sorts of undesirable phenomena can be observed… well-documented failures of this type are precisely the reason that the tuning of the step-size of SGD plays such an important role in the field.” We cannot agree to this without a reference. To be specific, it is true, as the reviewer claims, that it is well known and not surprising that the gradient explodes when the learning rate is too large, but, to the best of our knowledge, the four main discoveries of this work have not appeared or have been systematically studied in any previous work and should appear surprising to a general audience. Also, the fact that this is surprising is agreed upon by the other four reviewers.
> > * To be specific, our four main discoveries are: (1) SGD may converge to local maximum; (2) SGD may prefer sharper minima; (3) escaping efficiency may decrease with increasing learning rate; (4) AMSGrad may converge to local maxima. To the best of our knowledge, none of these deleterious phenomena have been “well-documented” by previous works in the context of machine learning.
> >
> > *Q2: "Now, what is interesting in the authors' analysis is the specification of the range of for which these deleterious outcomes can arise in the quadratic toy model of item (1) above and the CLT analysis that they provide (which, incidentally, relies heavily on the specific loss model considered). However, what lessons can be inferred for neural network training from this 1-dimensional example is highly debatable - and the authors' one-neuron, quadratic-activated "neural-network-like" example in Section 6.3 is no less artificial than the examples of Sections 4 and 5. [By the way, given that both of the authors' examples are custom-made to allow calculations in quasi-closed form, statements concerning "restrictive and unrealistic assumptions about the nature of the noise" seem out of place.]"*
> > * We agree that the neural network example is limited in complexity, but it is a significant advancement over the previous works on saddle points and indeed shows the limitation of the common previous assumptions. None of the previous works that study saddle points contain examples that can be regarded as a proper neural network (including the four references you provided) -- however, a textbook definition of a feedforward network would include our example as a proper instance of a neural network. We added this clarification to Section 6.3.

---

> > > ### Author Response · Authors · 2021-11-19
> > > **Reply part 3**
> > >
> > > *Q3: "What I found lacking in the above is a thorough comparison with the work of Ge et al. [1]. Specifically, how does the authors' "trapped regime" compare to that predicted by [1]? More concretely, if we take Corollary 1 as a starting point, what is the precise value for below which [1] guarantees saddle-point avoidance? Is there a gap with the authors' trapping bounds or are they sharp? Given that the two results – that of the authors' and that of [1] – indicate completely different behaviors for (SGD), I would expect a much more in-depth comparison between the two. In a similar vein, it is also unfortunate that the authors seem to ignore a series of results showing that SGD with a decreasing step-size avoids saddle-points with probability – cf. the general paper [2] and the more recent references [3,4] below. "
> > > * Thanks for raising this point. The difference in the theoretical prediction is a result of the assumptions. To be specific, the result of Ge et al. [1] assumes the following two things we do not assume: (1) the fluctuation of the gradient around its mean is upper bounded by a radius $Q$ (Def 3 in [1]); however, in our example, this upper bound does not exist; (2) Ge et al. also assumes that the loss function is a strict saddle, which implies that there are local minima existing close any saddle points (Def 5), which also does not apply to our example. Note that these two crucial assumptions are violated by the 1-neuron network example we studied in Section 6.3, and, therefore, the result of Ge et al. cannot be applied to our examples. We added this detailed discussion to Section 5.1.
> > > * Also, we added a discussion about the limitation of our work in light of the papers [2,3,4] you referred to. Indeed, our example only shows the possibility for a finite learning rate and does not rule out the possibility that a vanishing learning rate can help escaping the saddle points. See the footnote at the end of Section 5.1
> > > * The range of learning rate found in corollary 1 is actually sharp. The statement of corollary 1 can be stated as “if and only if”. The sharpness of this boundary is also numerically confirmed by Figure 1-Right.
> > >
> > > *Q4: In fact, one of the most popular neural network training methods involves staircase step-size schedules that begin at a large value which is then progressively halved towards the end of the training horizon - and this, precisely to enhance the algorithm's convergence properties. The authors' theory does not cover this important part of the literature, so their claims are - at best - incomplete in this regard.*
> > > * We agree that (and as we acknowledged in the manuscript) a limitation of our analysis is that we do not consider a decreasing learning rate, and this part is left as an important future work.
> > >
> > > *Q5: To summarize, the paper does not "run counter to the established wisdom of the field". If anything, it serves to reinforce this wisdom by providing an interesting cautionary tale to the effect that "SGD with a large, constant step-size can converge to local maxima".*
> > > * Thanks for raising this point. This is exactly our point. Our result does not disprove the established results in (and this is never claimed in the manuscript), for example, Ge et al. because the assumptions are different.  We have thus rewritten the sentences that involve “run counter,” both in the abstract and conclusion; for example, in the abstract, we now say “when not in the regimes that the previous works often assume, SGD can exhibit many strange behaviors that the established wisdom of the field cannot explain.”
> > > * However, we would like to comment that our result does suggest that some recent popular beliefs/conjectures about SGD need revision (if not incorrect), such as (1) SGD prefers flatter minima (for example, see Xie et al., 2021; Liu et al., 2021; Mori et al., 2021; Smith and Le, 2018; Wojtowytsch, 2021b we cited in the paper), (2) AMSGrad is convergent (see the original paper of AMSGrad), and (3) increasing learning rate helps escaping (see Kleinberg et al., 2018; or see the theorem 6 of Ge et al. 2015, where the convergence rate to a local minimum increase with an increased learning rate polynomially).

---

> > > > ### Author Response · Authors · 2021-11-19
> > > > **Reply part 4**
> > > >
> > > > *q6: "Title: see above. Without any further quantifiers (and despite the word "can"), the current title suggests that "convergence to maximizers" is a ubiquitous phenomenon - whereas it is anything but."*
> > > > * We disagree with this criticism. We believe that the word “can” is, in fact, the most important word in the title. Ignoring the most important word in a sentence and criticizing that it is wrong is logically incorrect.
> > > > * To be specific, the Oxford dictionary has the following definition of the word “can”: “Expressing objective possibility, opportunity, or absence of prohibitive conditions” Taking this definition, our title precisely states the following message: SGD has the possibility to converge to local maxima, which is what we have proved (showing existence by constructing an example). Therefore, according to the dictionary definition, our title is technically accurate and precise and does not suggest that the convergence to local maxima is ubiquitous to any proficient English speaker.
> > > >
> > > > *q7: Abstract: the statement concerning restrictive and unrealistic assumptions is out of place. If the authors refer to the "multiplicative" nature of the noise, they should note that the stochastic loss function $\hat{L}(w; x) = (w-x)^2/2, x\in-1, 1$, has additive noise at its minimum and it is neither more nor less "realistic" than their toy example. In fact, it can be argued that the principal reason that the author's example exhibits convergence to a non-minimizing stationary point is that all batches become critical at the same point - but, in turn, this is an unrealistic assumption in itself. At any rate, blanket statements like this should be avoided unless the authors are prepared to back them up with a deeper theoretical treatment.*
> > > > * We have updated the abstract to moderate our statement.
> > > >
> > > > *  However, the fact that our example is special does not imply that the previous theoretical works do not have restrictive and unrealistic assumptions when applied to deep learning. As we have shown in section 6.3 and appendix section A.5, there exist examples of neural networks that the previous theoretical works cannot explain (hence, the assumptions are “restrictive”) -- therefore, our claim is not wrong and, in fact, backed up by discussion and evidence.
> > > >
> > > > *q8: A more important issue: the $a>0$ case is poorly explained in the paper, and the corresponding part of Fig. 1 is misleading. For $a>0$, convergence to $0$ is the desired behavior, so coloring that region as "problematic" is not appropriate.*
> > > > * This is factually incorrect. In figure 1 right, as we have described in the paper, the color only describes the probability of not converging to the stationary point (white: probability 0, blue: probability 1) and has nothing to do with whether the behavior is “problematic” or not.
> > > > * The reason for not discussing the $a>0$ case is because it is not relevant for the main messages of this work and already well-understood in previous works: as we have pointed out in the paper (discussion below Eq. 2), see Ziyin et al. (2021), for example.
> > > >
> > > > *q9: I was bemused with the "sharp vs. shallow" statement: especially since the trace of the Hessian is only a local attribute that does not suffice to characterize the basin of attraction of a (non-quadratic) minimum, I do not see why this is "counter to established wisdom to the field". Again, I would recommend providing an accurate mathematical description and letting readers draw their individual conclusions instead of trying to maneuver the narrative in this way.*
> > > > * We agree that the trace (or the determinant) of the Hessian is only a local value and may not faithfully reflect the local sharpness. However, it is widely regarded as one of the main measures of the sharpness of a local minimum (along with the determinant of the local Hessian). Mathematically, this is justified because, within a second-order approximation, the trace indeed reflects the volume of the local minimum. For example, see Dinh et al. (2017) and Xie et al. (2020) for its usage. Therefore, our usage of the word sharpness is consistent with contemporary literature and should not mislead the audience familiar with the relevant works. Lastly, we comment that it should be interesting and important to explore better ways of defining the sharpness of a local minimum but beyond the scope of the present work.
> > > > * Also, we point out that we do not discuss “sharp vs. shallow” but “sharp vs. flat.” The word “shallow” is an inaccurate description of our discussions in Section 5.3

---

> > > > > ### Author Response · Authors · 2021-11-19
> > > > > **Reply part 5**
> > > > >
> > > > > *q10: The authors are making a series of claims for the applicability of their results to neural networks, but these are based on a "network" with a single neuron. I would again recommend moderation - otherwise, this is a textbook case of a faulty generalization.*
> > > > > * We moderated our phrasing and added a note on the limitation of our neural network example to the end of Section 6.3.
> > > > > * However, we stress that we did not argue that this is a generalizable example -- all we have argued is that there is such a possibility. For example, the first question in our conclusion section directly questions whether the example we presented is generalizable: “Do the behaviors we realize in simple constructions also occur in realistic deep networks?” Also, at the beginning of the neural network example, we also stated “...we construct a neural-net-like optimization problem to show that our zoo of odd effects might indeed arise...” The word “might” is sufficient to convey that the statement only suggests the existence of a possibility and may or may not be generalizable. Even in the original draft, it is not the intention of the authors to argue for the generalizability of the result, and we regard the generalizability of this example as one of the most important future steps.
> > > > >
> > > > > *q11: The statement that "SGD noise is multiplicative and state-dependent" is too vague and lacks context.*
> > > > > * See “note on the relevance of the continuous-time analysis and multiplicative noise” at the top of this reply.
> > > > >
> > > > > *q12: When mentioning continuous-time approaches, I was surprised that the authors did not discuss the stohastic approximation framework for the study of SGD by Ljung, Kushner and Yin, and Pemantle (to state but some of the most classical results). I already mentioned this literature in the context of SGD with a decreasing step-size above, and since the authors seem to be interested in the continuous-time limit, this is a second reason why this literature should be discussed in detail. This also comes up when discussing related works in p.2 (as these works do not assume "artificially injected noise").*
> > > > > * It is important that we refer to the works that are actually relevant. However, your reference is vague and too general for us to make an update. We looked up the works of Ljung, Kushner and Yin, and Pemantle and could not find one that is specifically relevant. We will add a reference and discussion if you can point out what results in which references are particularly relevant.
> > > > >
> > > > > *q13: Still on the issue of related work: the authors seem to be confusing the Polyak-Łojasiewicz condition with the Kurdyka-Łojasiewicz condition. The former is global, and indeed restrictive; the latter is local, and includes all semi-algebraic functions (or more general any function defined by an o-minimal structure). I would encourage the authors to study in more depth the work of Bolte and co-authors on the topic - and, in addition, I would like to point out that all examples considered by the authors satisfy the KL condition.*
> > > > > * This criticism lacks evidence. As in the relevant previous works (for example, Karimi et al., 2020;Wojtowytsch, 2021a; Vaswani et al., 2019), this work only discussed the Polyak-Łojasiewicz condition, not the Kurdyka-Łojasiewicz condition. We thus do not find any ambiguity in the paper about this condition (we also clearly defined the PL condition in the appendix section A.5).
> > > > >
> > > > > *q14: Appendices B and C are quite disconnected from the rest of the paper: they concern a completely different model with very different results and attributes, so I would suggest removing them altogether.*
> > > > > * This criticism is factually incorrect. The appendices B and C study the same model as is used in the main text. To be specific, Section C.2-C.4 contains discussions and analyses that are highly relevant for the example in Section 5.1 (and C.4 is exactly the same model as in Section 5.1).
> > > > > * Again, see “note on the relevance of the continuous-time analysis and multiplicative noise” at the top of this reply. Understanding these two sections can be essential for some readers to understand the results of our work.

---

> > > > > > ### Author Response · Authors · 2021-11-19
> > > > > > **Reply part 6**
> > > > > >
> > > > > > *q15: While interesting and easy to work out, Proposition 1 is very special: if $\lambda=1$, we get an exact Newton update, which gets to $0$ in a single iteration, and thus remains there forever due to the authors' multiplicative randomness model. In this regard, Proposition 1 is not truly representative of what's going on (e.g., in Proposition 2).*
> > > > > > * This is debatable and really depends on the perspective. From the perspective of the types of convergence, the SGD mechanism behind proposition 1 may be different from that of proposition 2. However, from the perspective of continuous-time analysis, there is really the same mechanism that is governing the SGD dynamics behind propositions 1 and 2 (namely, the fact that the noise has dominated the gradient). We added a note to reflect two possible different perspectives at the end of Section C.4.
> > > > > >
> > > > > > *q16: In their discussion of Proposition 1, the authors also briefly discuss the difference between convergence in probability and convergence in expectation. I believe this distinction in the modes of convergence of random variables is an important take-away of this work, and one worth describing in more detail. If the paper's message is that we need to be careful about how we interpret SGD convergence and avoidance results, then this should be made clearer.*
> > > > > > * This is indeed an important message of the paper (as also pointed out by reviewer *MX9i*), but we feel that the other insights we have offered are no less important than this particular message, and the present length of this discussion reflects the balance we decided to take in order to also sufficiently discuss the other messages.

---

> ### Comment · Reviewer_DCLz · 2021-11-22
> **Thanks for the extensive reply, but my criticisms stand as stated (Part 1/2)**
>
> I thank the authors for their replies. Given the number of papers that I am currently treating, I do not have the time to reply to every individual point of the authors' rebuttal, so I did my best below to factor the relevant points accordingly.
>
> 1. **Evaluation of the revision:**
> Overall, I do not find that the revised version of the paper reflects any of my criticisms in any substantive manner. In particular, the *actual* revision of the manuscript was minimal to nonexistent (one phrase in the first four sections, two paragraphs plus a footnote in p. 5, and one phrase in the last section). Also, despite the length of the authors' rebuttal, the essence of my concerns was not addressed, so I am not comfortable in changing my score or recommendation (nor have the authors' replies convinced me that my criticisms are "due to [my] misunderstanding of the paper's theory").
>
> 1. **On the suprise factor:**
> First, to avoid any misunderstandings here, I would like to reiterate that I do not share the viewpoint that something needs to be "surprising" in order to be worthy of publication.
>   With this caveat out of the way, what I said is that "if the step-size is not chosen appropriately, all sorts of undesirable phenomena can be observed (both in practice and in theory)." Escape to infinity is one such phenomenon; another is chaotic trajectories, which occur even in the case of *deterministic* gradient descent with a convex potential (the authors can check the papers of Piliouras and co-authors for this); etc.
>   The authors show a different failure that can occur when the step-size of a gradient method is very large. I agree that this is mathematically interesting (I said as much in my original review), but I do not agree that this goes against the "established wisdom of the field" or that "it is the direct opposite of that of many related works". As I said in my original review, I believe that a proper revision would require a significant rewrite from the authors -- but since the actual changes provided were minimal, I maintain my original assessment.
>
> 1. **The continuous-time analysis:**
> The authors claim that "*the indispensable role of continuous-time analysis is also neglected in [my] review*". To be precise, what I said was that the continuous-time analysis is disconnected from the discrete-time one, and this because the exact connection between the discrete- and continuous-time regimes is glossed over by the authors. [I am referring here to the precise way that the diffusion (30) approximates SGD, the lack of discussion regarding the fact that the variance of the Itô process would have to grow as a square root of the step-size in order to make this approximation valid, the lack of precision as to whether this concerns a finite-horizon or asymptotic comparison, etc.]
>   More importantly, if this analysis is so "indispensable" as the authors claim in their rebuttal, it should have been included in the original paper, so that it could be properly reviewed and evaluated on its proper merits - the authors can't have their cake and eat it, too.
>
> 1. **The multiplicative nature of the noise:**
> It is misleading to state that "*the multiplicative dependence is a direct consequence of the minibatch sampling technique*"; it is a consequence of the minibatch sampling technique ***and*** the fact that each summand shares the same critical set. The "*and*" here is crucial, and it is precisely the reason that I gave the example of $\hat L(w;x) = (w-x)^2/2$, $x\in\{-1,1\}$ in my review: in this case, minibatch sampling **does not** give multiplicative noise at the critical point of $\mathbb{E}[\hat L]$, so the authors' statement above is unjustified.
>
> 1. **The relation with Ge et al :**
> The authors claim that the main differences with the work of Ge et al. is (a) that the noise does not have finite support; and (b) that the "strict saddle" property is not satisfied. This is false: Definition 5 of Ge et al. states that a function has the "strict saddle" property if, for any point $w$ in the function's domain, *at least* one of the stated properties is satisfied. One of these properties is that the Hessian have a negative eigenvalue, and this is trivially true for the function $a w^2/2$, $a<0$, considered by the authors in Proposition 1.
>   So, no, the reason that the theory of Ge et al. does not apply is not that the maximum is not a strict saddle, but the fact that SGD is run with a very large step-size. This was also the reason that I asked the authors to compare their step-size bounds with the bounds of Ge et al. -- but, regrettably, they did not address this point.
>
> [Continued in a second post]

---

> > ### Comment · Reviewer_DCLz · 2021-11-22
> > **Thanks for the extensive reply, but my criticisms stand as stated (Part 2/2)**
> >
> > [Continued from previous post]
> >
> > 6. **The case of a decreasing step-size:**
> > The authors included a footnote in p.~5, which does not treat the question in any subtantive manner. As with the case of the work of Ge et al., this should have been moved and discussed at length in the related work section.
> >   The authors also ask about more precise pointers regarding the papers that I mentioned in my original review regarding the decreasing step-size case. So be it: Ljung and Kushner & Yin provide the precise link between continuous- and discrete-time versions of SGD (which the author seem to be very interested in); Pemantle provides the first result showing that stochastic approximation algorithms (like SGD) avoid critical points that are unstable for the underlying dynamics (hyperbolic saddles in the case of non-convex optimization); the more recent works by Vlaski & Sayed and Mertikopoulos et al. provide a series of finer results for SGD and provide precise conditions under which saddle points are avoided, with probability $1$ or in expectation.
> >
> > 1. **On the title:**
> > The issue here is simple: if the authors wish to have an informative title, then they should include the relevant quantifiers under which SGD "can" converge to maximizers. The point of theory papers is to inform the reader, not to make statements whose interpretation is left to the eye of the beholder. Otherwise, the authors' title is as informative as saying that "gradient descent can converge in a single iteration": this "can" happen but, without specifying the conditions under which this _does_ happen, this is a misleading statement.
> >
> > 1. **On the neural network example:**
> > The authors state in their revision that "*while this example is a highly limited special example, it is closer to actual deep learning than the popular settings of escaping saddle points with SGD*". It is indeed arguable if the noise conditions of Ge et al. are satisfied in DL models (but, at the same time, even if they aren't, it is also debatable whether this failure is crucial or not). By contrast, I am not aware of *any* deep learning model with 2 parameters, so I frankly fail to see in which way the authors' example is _closer_ to DL than that of Ge et al.
> >
> > ---
> >
> > In summary, I regret to say that my criticisms stand as stated: while I do believe there is merit in the authors' _mathematical_ results, the paper contains a series of statements and claims that range from the (often) ambiguous to the (sometimes) misleading. The authors' revision and rebuttal did very little to assuage my concerns, so I maintain my original score and assessment.

---

> ### Comment · Reviewer_DCLz · 2021-11-22
> **Second reply to the authors (1/2)**
>
> I just finished going through the authors' second set of replies. In the interest of giving some constructive input to the authors despite the shortage of time, I reply below to the the three points that I found most representative of the overall discussion.
>
> 1. **The continuous-time analysis:**
> The authors claim that this is an "indispensable" part of the paper and they further assert that the lack of a precise link between the discrete- and continuous-time systems is not a relevant criticism "because it is not the scope of this work to study whether this approximation is valid". I perused the works cited by the authors as motivation for the continuous-time setting and I couldn't find any precise link between the invariant measure of SGD and that of (30) (the solution of the associated FP equation), nor any asymptotic stochastic approximation result linking the two frameworks. If there is one to be found, I invite the authors to state it precisely so that it can be evaluated and assessed as a precise mathematical statement. Otherwise, if there is no such link, I fail to see how the continuous-time analysis they provide is "indispensable" for understanding SGD or why "in continuous time, SGD is equivalent to (23)", etc. [There is also a point of order to be made here – namely that supplemental material is optional – but I am willing to let this slide]
>
>     Incidentally, for Kushner and Yin: the first google hit for "Kushner Yin" is their seminal book "*Stochastic approximation algorithms and applications*". This is one the main references for linking discrete- and continuous-time systems, and the authors may want to check out this literature in detail.
>
> 1. **Writing style:**
> Perhaps the most representative example here is the title. The current title "SGD can converge to local maxima" is factually correct. The title I suggested in my review "SGD with a large, untuned step-size can converge to local maxima" is also factually correct. The difference is that the latter carries more information than the former, so it provides a more precise description of the authors' results and one which is less prone to misinterpretations.
>
>     Do I have any specific attachment to this title in particular? Absolutely not, this is not my point, I am only stating it as an example. My main criticism is that a theory paper should focus on precise mathematical statements, not vague sound bites (and factual correctness does not exclude vagueness).
>
>
> 1. **Minibatch sampling and multiplicative noise:**
> In their rebuttal, when replying to my point on the gradient noise at a critical point, the authors stated that "the multiplicative dependence is a direct consequence of the minibatch sampling technique". I replied that this is a consequence of minibatch sampling *and* the fact that each summand shares the same critical set, and I gave the function $L(w;x) = (x-w)^2/2$ as a simple example to illustrate my point. The authors missed this point in their reply: they rebutted that they "use the word "multiplicative noise" interchangeably with the word "state-dependent noise" [i.e.,] any noise whose variance is not constant in $w$" and they then proceeded to say that this example "is, unfortunately, irrelevant for deep learning because it does not contain a state-dependent noise".
>
>     Setting issues of terminology aside for the moment (I come back to this below), the noise being state-dependent has nothing to do with its variance vanishing at a critical point. To illustrate this, consider the following slightly less symmetric example than the one I provided in my original review: let $x$ be a Bernoulli variable taking the values $1$ and $2$ with equal probability, and let $\hat L(w;1) = 2(w-1)^2$ and $\hat L(w;2) = (w+2)^2$. If my back-of-the-envelope calculation is correct, the associated mean loss function is $L(w) = 3w^2/2 + 3$ and it is minimized at $w=0$. The variance of the gradient noise is $(w-4)^2$ so the noise is state-dependent; however, the gradient noise **does not** vanish at $w=0$, even though $L'(0)=0$ (and thus, in particular, it is not "multiplicative"). Thus, minibatch sampling does not imply multiplicative noise in this simple example.
>
>     Now, since the gradient noise induced by minibatch sampling does not necessarily vanish at a critical point, the authors' statement that "the multiplicative dependence is a direct consequence of the minibatch sampling technique" is, at best, confusing. And since understanding the precise mechanism that allows SGD to converge to maximizers is what this paper is about, the whole premise suffers.
>
>
> [Continued below]

---

> > ### Comment · Reviewer_DCLz · 2021-11-22
> > **Second reply to the authors (2/2)**
> >
> > [Continued from previous post]
> >
> > 3. **Minbatch sampling, continued:**
> > In the interest of moving forward and being as constructive as possible, let me submit a precise question to the authors: *can you provide a loss function $\hat L(w;x)$ such that $L(w) = \mathbb{E}[\hat L(w;x)]$ is maximized at $w=0$ (as per Proposition 1), the gradient noise has non-zero variance at $w=0$, and SGD converges to 0 almost surely?*
> >
> >    If not, I maintain my original assessment that SGD cannot converge to a maximizer unless the noise vanishes at said point: this is the reason that the authors observe the behavior that they do, not minibatch sampling or some link to deep learning - and this paints a completely different picture of what's going on.
> >
> > ---
> >
> > In summary, I believe that the authors are treating an important topic, one which merits a clean and precise treatment, with no claims that are open to vagueness or interpretation. However, after this lengthy exchange, I am even more convinced that the (important) questions that the authors set out to answer are not addressed at sufficient depth or clarity, so my original score and assessment remain unchanged.

---

> > > ### Author Response · Authors · 2021-11-23
> > > **Third response (part 1)**
> > >
> > > Thanks for your prompt update. We do find your updated review constructive, and this also made us realize that we have misunderstood part of your original points. We actually removed our second response before your second reply because we start to feel that our original presentation does require a lot of revision that you suggested even before your second reply. In particular, our original presentation does contain a lot of points/claims that should have been stated more carefully or completely left to the judgment of the readers (and not stated at all). Misleading the readers is not what we intend, and we make a major revision of the draft in this light.
> > >
> > > To summarize, we made the following changes and are curious about your opinion (both the revision from this time and the first time are colored in orange). A lot of potentially misleading claims/statements are also removed, but this cannot be highlighted, unfortunately. In particular (see the newly updated manuscript),
> > > 1. We propose to change the title to make it more informative. Your suggested title is indeed accurate and informative in describing our result, and we adopt this title: "SGD with A Large and Untuned Learning Rate Can Converge to Local Maxima"
> > > 2. We removed all the statements about surprisingness. We agree that this is a subjective statement and should be completely left to the judgment of the readers
> > > 3. We agree with your suggestion that our result does not go against the conventional wisdom and may, in fact, reinforce them (namely, that a carefully chosen and scheduled learning rate can help convergence and escaping saddle points). This is reflected in our updated abstract, the related works section, and the conclusion. For example, in the abstract, we state "One crucial message that our work delivers is that the learning rate needs to be chosen and scheduled carefully to guarantee convergence." In the related works section, we now state "It is important to note that our result does not contradict these previous results. To the opposite, our result may be interpreted as reinforcing the crucial messages in these previous works, namely, that the learning rate needs to be chosen and scheduled carefully to guarantee convergence."
> > > 4. We removed all claims that the neural network example we offered is more realistic than the previous works. We realized that this should be a point that is left to the judgment of the readers. We also added a cautionary note at the end of section 6.3: "Finally, we add the cautionary note that it remains an open question whether this toy example we offer is relevant for deep learning in practical settings or not, and it should be an important future work to investigate this problem."
> > > 5. The related works by Pemantle, Vlaski, and Mertikopoulos are now discussed in more detail in the related works section. In addition, this discussion is used for arguing that our work does not go against conventional wisdom
> > > 6. At various places, we mentioned that the example we proposed is a special example, and the main mechanism behind it is the state-dependent nature of the noise __and__ that the data points share the same critical set, as you suggested. In particular, we state in section 5.1: "One special feature of this example is that the state-dependent noise vanishes at the saddle point. In this particular example, this is achieved by having an SDG noise proportional to the gradient thus vanishing at the saddle point. However, in general, having vanishing noise and state-dependent noise is not always related. Thus, what this example provides is a worst-case scenario: the state-dependent noise leads to vanishing noise."
> > > 7. For the same reason, we also removed statements that claim relevance for deep learning. This is also a point that we should leave to the reader to judge. At one place we do discuss whether our result is deep learning-relevant or not, we state clearly: "While our example may or may not be directly relevant for realistic settings in the  deep learning practice, our work highlights the importance of analyzing the actual noise structure imposed by a deep neural network in future works."
> > > 8. In section 4 (the warmup example section), we added a little to discuss convergence in expectation vs. convergence in probability as suggested in your initial review
> > > 9. The Ge et al. discussion is expanded a little more to make the discussion more informative
> > > 10. In the conclusion, we clearly state the limitation of this work: "The limitation of our work is clear. At best, all the constructions we made are minimal and simplistic toy examples that are far from practice, and investigating whether the discovered messages are relevant for deep learning or not is the one important immediate future step." Moreover, we carefully removed all the claims that contradict (or seem to contradict) this statement of limitation
> > > 11. We also removed the claims that the previous works are restricted.

---

> > > > ### Author Response · Authors · 2021-11-23
> > > > **Third response part 2**
> > > >
> > > > Please feel free to point out any remaining statements that are potentially misleading, we are willing to make the relevant revisions in future versions.
> > > >
> > > > The following answers the specific questions you raised in the second reply.
> > > >
> > > > **Title**: As commented above, we have updated the manuscript to reflect your suggestion. We feel that the updated title is indeed more informative. We also updated the relevant claims that features a similar misleading problem in our original manuscript, as described by the first part of the third response
> > > >
> > > > **Multiplicative noise**: Thanks for the explanation. We agree to your comment in the first reply that “this is a consequence of minibatch sampling and the fact that each summand shares the same critical set,” and this clarification is now added to the manuscript. We emphasized that our example is special in the sense that the noise vanishes at the critical point. Therefore, it is also incorrect for us to claim in our first response that such a phenomenon is a “direct consequence of minibatch sampling.”
> > > > As a specific answer to the question you submitted, we see your point and are unable to construct such an example.
> > > >
> > > > **The continuous-time analysis**: Thanks for the explanation. Our statement about “indispensable” may be too strong -- please let us explain what this section is for and why we feel that it is sufficiently justified for its purpose. The goal for the continuous-time analysis we offered is to offer more qualitative insight into what might have been the cause for the phenomenon in the special example we proposed, and we do not claim it to be a technical contribution.
> > > >
> > > > It is true that the previous works we referred to do not show the direct connection between the invariant distribution of the discrete-time SGD and the continuous-time SGD, but this does not imply that our original discussion is completely unjustified. In particular, the logic we take is simple and exists in the works we cited: treat SGD as a discretization of the continuous version, impose an integration scheme, and apply the stationary Fokker-Planck formalism. The justification for our discussion the continuous-time analysis is empirical. In particular, the agreement between the prediction of the continuous time analysis and numerical simulation is great when the learning rate and the local curvature $a$ is small (see the updated discussion relating to Figure 9 in section C.4 for the empirical evidence), and this suggests that the used continuous-time analysis is sufficient for analyzing this problem in the regime where the continuous-time approximation is valid.
> > > >
> > > > In summary, the goal of the continuous-time analysis is to help with understanding the problem, and its limitations can be clearly judged by a mathematically mature reader. Furthermore, since this part does not constitute the main contributions of the work, we also feel that this may not constitute a reason for rejection.

---

> > > > > ### Comment · Reviewer_DCLz · 2021-11-24
> > > > > **Thanks for the replies**
> > > > >
> > > > > Thank you for your replies. I went through the new revision of the manuscript in its entirety, and all my major concerns were addressed, so I changed my score accordingly.

---

> ### Comment · Reviewer_DCLz · 2021-11-24
> **Reassessment after second revision**
>
> I thank the authors for their reply and their second revision. I did not reply immediately, first because of a shortage of time, and second because I wanted to take some time off, and then re-assess everything from the beginning with a fresh pair of eyes.
>
> All my major concerns were addressed in this revision and I am happy to recommend acceptance as a result. In particular, the authors' statement that "this work can be seen as a worst-case ana[l]ysis of SGD" (note the small typo btw) is, in my opinion, a very precise and concise encapsulation of the paper. Of course, determining whether these worst-case occurrences are actually "common" or "anomalous" is not an easy task, and I believe it can be an important direction for future research in the ML community.
>
> Some minor bugs remain (to be expected given the tight turnaround time), but nothing that can't be fixed with a calm proof-reading.
>
> A clarification: my score should be interpreted as a "7" (accept) but ICLR doesn't have this granularity.

---

> > ### Author Response · Authors · 2021-11-25
> > **reply**
> >
> > Thanks for your update and for your careful review.
> >
> > We will proofread and revise our manuscript more carefully in a future version of the draft.

---

### Author Response · Authors · 2021-11-19
**Rebuttal Summary  (UPDATED)**

Note: This thread is updated after the third revision.
Again, we thank all the suggestions made by the individual reviewers. In addition to the revisions made in the first revision, we made the following revisions to reduce the overclaiming issues:

1. We propose to change the title to make it more informative. Your suggested title is indeed accurate and informative in describing our result, and we adopt this title: "SGD with A Large and Untuned Learning Rate Can Converge to Local Maxima"
2. We removed all the statements about surprisingness. This is a subjective statement and should be completely left to the judgment of the readers
3. Our result does not go against the conventional wisdom and may, in fact, reinforce them (namely, that a carefully chosen and scheduled learning rate can help convergence and escaping saddle points). This is reflected in our updated abstract, the related works section, and the conclusion.
4. We removed all claims that the neural network example we offered is more realistic than the previous works. We realized that this should be a point that is left to the judgment of the readers.
5. The related works by Pemantle, Vlaski, and Mertikopoulos are now discussed in more detail in the related works section. In addition, this discussion is used for arguing that our work does not go against conventional wisdom
6. At various places, we mentioned that the example we proposed is a special example, and the main mechanism behind it is the state-dependent nature of the noise __and__ that the data points share the same critical set. Thus, what this example provides is a worst-case scenario: the state-dependent noise leads to vanishing noise."
7. For the same reason, we also removed statements that claim relevance for deep learning.
8. In section 4 (the warmup example section), we added a little to discuss convergence in expectation vs. convergence in probability as suggested in your initial review
9. The Ge et al. discussion is expanded a little more to make the discussion more informative
10. In the conclusion, we clearly state the limitation of this work: "The limitation of our work is clear. At best, all the constructions we made are minimal and simplistic toy examples that are far from practice, and investigating whether the discovered messages are relevant for deep learning or not is the one important immediate future step." Moreover, we carefully removed all the claims that contradict (or seem to contradict) this statement of limitation
11. We also removed the claims that the previous works are restricted.



Below is the summary of the first revision
--------------
We thank the reviewers for finding value in our results.

 We have updated the manuscript to reflect the constructive suggestions by the reviewers.  The main revisions are colored in orange, and we invite the reviewers to skim through our updates and give further feedback if there are any. In particular, the major revisions include:
1. Combination of Theorem 1 and proposition 3 to reduce the redundancy of Section 5.2, as suggested by reviewer *bMKE*
2. Additional discussion in section 6.3 to stress that the neural network example we proposed may be a special case and may not be generalizable to more realistic architectures and tasks, as suggested by reviewer *DCLz*
3. Additional discussion of the difference of the present work with that of Ge et al., at the end of Section 5.1, as suggested by reviewer *DCLz*
4. Addition of discussion regarding the limitation of this work in light of the three papers pointed out by *DCLz*, which study the escaping behavior of SGD/GD with a decreasing learning rate.  See the footnote at the end of Section 5.1
5. Clarification of the essential role of the multiplicative noise of SGD in our example at the end of Section 5.1 and Appendix Section C.4, in response to reviewer *DCLz*
6. Toning down many claims as suggested by *DCLz*, mainly in the abstract, conclusion, and introduction

After this revision, we are confident that the discussions in the present manuscript are more precise, accurate, and concise.
We reply to the constructive questions of the reviewers below.

---

### Decision · Program_Chairs · 2022-01-20

**Decision:**

Accept (Spotlight)

**Comment:**

Overall, the paper provides interesting counter examples for the SGD with constant step-size (that relies on a relative noise model that diminishes at the critical points), which provide critical (counter) insights into what we consider as good convergence metrics, such as expected norm of the gradient.

The initial submission took a controversial position between the mathematical statements and the presentation of the statements on the behavior of the SGD method in non-convex optimization problems. While the mathematical is sufficient for acceptance at ICLR, the presentation was inferring conclusions that could have been misread by the community.

I am really happy to state that the review as well as the rebuttal processes helped improved the presentation of the results that I am excited to recommend acceptance.